# Human noise blindness drives suboptimal cognitive inference

Santiago Herce Castañón[1,2], Rani Moran [3,4], Jacqueline Ding[1], Tobias Egner[5,6], Dan Bang [3] &
Christopher Summerfield[1]

Humans typically make near-optimal sensorimotor judgements but show systematic biases when making more cognitive judgements. Here we test the hypothesis that, while humans are sensitive to the noise present during early sensory encoding, the "optimality gap" arises because they are blind to noise introduced by later cognitive integration of variable or discordant pieces of information. In six psychophysical experiments, human observers judged the average orientation of an array of contrast gratings. We varied the stimulus contrast (encoding noise) and orientation variability (integration noise) of the array. Participants adapted near-optimally to changes in encoding noise, but, under increased integration noise, displayed a range of suboptimal behaviours: they ignored stimulus base rates, reported excessive confidence in their choices, and refrained from opting out of objectively difficult trials. These overconfident behaviours were captured by a Bayesian model blind to integration noise. Our study provides a computationally grounded explanation of human suboptimal cognitive inference.

[1] Department of Experimental Psychology, University of Oxford, Oxford OX1 3UD, UK. [2] Department of Psychology and Educational Sciences, University of Geneva, 1205 Geneva, Switzerland. [3] Wellcome Centre for Human Neuroimaging, University College London, London WC1N 3BG, UK. [4] Max Planck UCL Centre for Computational Psychiatry and Ageing Research, London WC1B 5EH, UK. [5] Center for Cognitive Neuroscience, Duke University, Durham, NC 27710, USA. [6] Department of Psychology and Neuroscience, Duke University, Durham, NC 27708-0086, USA. These authors contributed equally: Dan Bang, Christopher Summerfield. Correspondence and requests for materials should be addressed to S.H.C. (email: s.hercecastanon@gmail.com)

The question of whether humans make optimal choices has received considerable attention in the neural, cognitive and behavioural sciences. On the one hand, the general consensus in sensory psychophysics and sensorimotor neuroscience is that choices are near-optimal. For example, humans have been shown to combine different sources of stimulus information in a statistically near-optimal manner, weighting each source by its reliability[1–6]. Humans have also been shown to near-optimally utilise knowledge about stimulus base rates to resolve stimulus ambiguity[7–11].

On the other hand, psychologists and behavioural economists, studying more cognitive judgements, have argued that human choices are suboptimal[12]. For example, when required to guess a person's occupation, humans neglect the base rate of different professions and solely rely on the person's description provided by the experimenter. Such suboptimality has been attributed to insufficient past experience[13], limited stakes in laboratory settings[14], the format in which problems are posed[15], distortions in representations of values and probabilities[16], and/or a reluctance to employ costly cognitive resources[17,18]. However, an account of human decision-making that can explain both perceptual optimality and cognitive suboptimality has yet to emerge[19].

Here we propose that resolving this apparent paradox requires recognizing that perceptual and cognitive choices often are corrupted by different sources of noise. More specifically, choices in perceptual and cognitive tasks tend to be corrupted by noise that arises at different stages of the information processing leading up to a choice[20–23]. In perceptual tasks, experimenters typically manipulate noise arising before or during sensory encoding. For example, they may vary the contrast of a grating, or the net motion energy in a random dot kinematogram, which affects the signal-to-noise ratio of the encoded stimulus and in turn the sensory percept[24]. Conversely, in cognitive tasks, which often involve written materials or clearly perceptible stimuli, experimenters typically seek to manipulate noise arising after stimulus encoding. For example, they may vary the discrepancy between different pieces of information bearing on a choice, such as the relative costs and benefits of a consumer product[18]. These types of judgement are difficult because they require integration of multiple, sometimes highly discordant, pieces of information within a limited-capacity cognitive system[25–27].

Here we test the hypothesis that, while humans are sensitive to noise arising during early sensory encoding, they are blind to the additional noise introduced by their own cognitive system when integrating variable pieces of information. We tested this hypothesis using a novel psychophysical paradigm that separates, within a single task, these two types of noise. In particular, observers were asked to categorise the average tilt of an array of gratings. We manipulated encoding noise (i.e. the perceptual difficulty of encoding an individual piece of information) by changing the contrast of the array of gratings, with decisions being harder for low-contrast arrays. Second, we manipulated integration noise (i.e. the cognitive difficulty of integrating multiple pieces of information) by changing the variability of the orientations of individual gratings, with decisions being harder for high-variability arrays. Manipulating these different sources of noise within a single task allows us to rule out previous explanations of the optimality gap which hinge on task differences. To pre-empt our results, we show that, while observers adapt near-optimally to increases in encoding noise, they fail to adapt to increases in integration noise. We argue that such noise blindness is a major driver of suboptimal cognitive inference and may explain the gap in optimality between perceptual and cognitive judgements.

## Results

### Experimental dissociation of encoding and integration noise.
All six experiments were based on the same psychophysical task

(see Methods). On each trial, participants were presented with eight tilted gratings organized in a circular array. Participants were required to categorise the average orientation of the array as oriented clockwise (CW) or counter-clockwise (CCW) relative to the horizontal axis (Fig. 1a, b). After having made a response, participants received categorical feedback about choice accuracy, before continuing to the next trial. We manipulated two features of the stimulus array to dissociate encoding noise and integration noise: the contrast of the gratings (root mean square contrast, rmc: {0.15, 0.6}), which affects encoding noise, and the variability of the gratings' orientations (standard deviation of orientations, std: {0°, 4°, 10°}), which affects integration noise. The underlying distribution of average orientations was identical for all experimental conditions.

In Experiments 1 ($n = 20$) and 2 ($n = 20$), we assessed the effects of contrast and variability on choice accuracy and evaluated participants' awareness of these effects. In both experiments, at the beginning of a trial, we provided a prior cue which, on half of the trials, signalled the correct stimulus category with 75% probability (henceforth biased trials), and, on the other half of trials, provided no information about the stimulus category (henceforth neutral trials) (Fig. 1b). The neutral trials provided us with a baseline measure of participants' choice accuracy in the different conditions of our factorial design, and the biased trials allowed us to assess the degree to which—if at all—participants compensated for reduced choice accuracy in a given experimental condition by relying more on the prior cue. In Experiment 2, to provide additional insight into participants' awareness of their own performance, we asked participants to report confidence in their choices (i.e. the probability that a choice is correct; Fig. 1c).

### Matched performance under encoding and integration noise.
We first used the neutral trials to benchmark the effects of contrast and variability on choice accuracy. As intended, choice accuracy decreased with lower contrast (ANOVAs; Exp1: $F_{(1,19)} = 15.5$, $p < 0.001$; Exp2: $F_{(1,19)} = 40.7$, $p < 0.001$; collapsed: $F_{(1,39)} = 49.1$, $p < 0.001$) and with higher variability (ANOVAs; Exp1: $F_{(1.2,24.5)} = 8.4$, $p < 0.01$; Exp2: $F_{(1.6,32.0)} = 26.4$, $p < 0.001$; collapsed: $F_{(1.4,56.9)} = 30.6$, $p < 0.001$). Our factorial design contained three critical conditions which allowed us to compare participants' behaviour under distinct sources of noise: (i) baseline, (ii) low-c and (iii) high-v. In the baseline condition, the total amount of noise is lowest (high contrast, 0.6; zero variability, 0°). In the low-c condition (low contrast, 0.15; zero variability, 0°), encoding noise is high but integration noise is low. Conversely, in the high-v condition, integration noise is high but encoding noise is low (high contrast, 0.6; high variability, 10°). As expected, choice accuracy was reduced both in the low-c and in the high-v conditions (about 12%) compared to the baseline condition ($t$-tests; baseline > low-c: $t_{(39)} = 9.24$, $p < 0.001$; baseline > high-v: $t_{(39)} = 9.69$, $p < 0.001$; Fig. 2a). Critically, choice accuracy was at statistically similar levels in the low-c and the high-v conditions ($t$-tests; Exp1, high-v > low-c: $t_{(19)} = 0.09$, $p > 0.9$; Exp2, high-v > low-c: $t_{(19)} = 0.31$, $p > 0.7$; collapsed, high-v > low-c: $t_{(39)} = 0.28$, $p > 0.7$; Fig. 2a). Overall, the results show that we successfully manipulated noise at different stages of information processing.

### Cue usage under encoding and integration noise.
We next leveraged the biased trials to assess the degree to which participants adapted to the changes in choice accuracy induced by our factorial design. Given the above results, we would expect participants to rely more on the prior cue in the low-c and the high-v condition than in the baseline condition. To test this prediction,

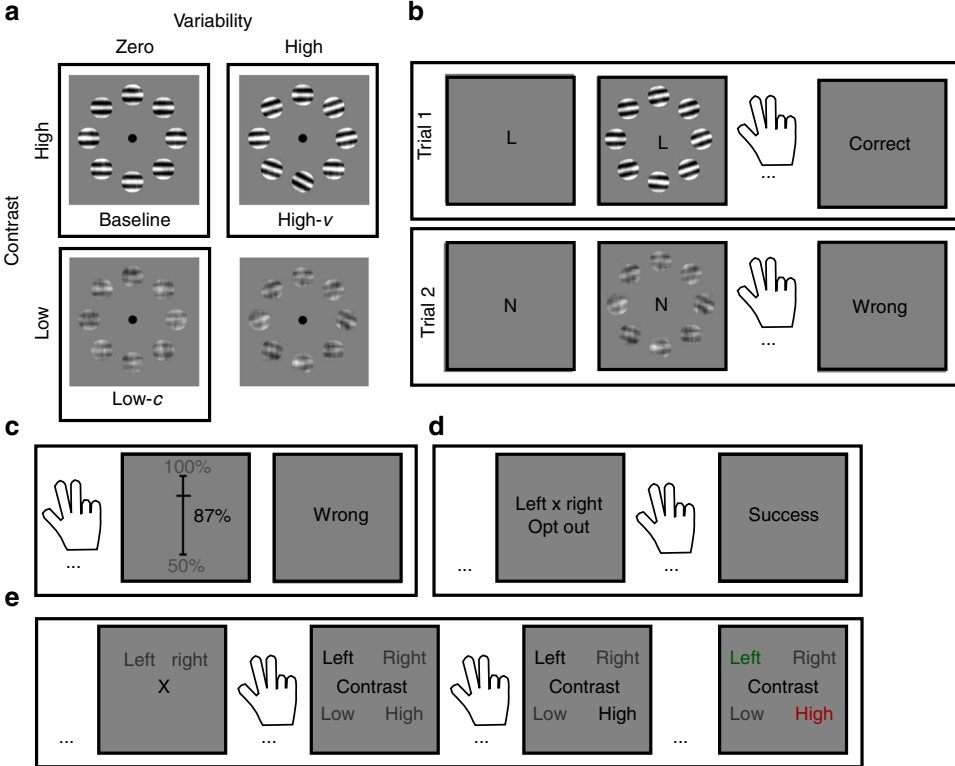

**Fig. 1** Experimental paradigm. **a** We manipulated the stimulus contrast and the orientation variability of an array of eight gratings in a factorial manner. The boxes highlight the three critical conditions. **b** Participants categorized the average orientation of an array as clockwise (CW, "left") or counter-clockwise (CCW, "right") relative to horizontal. A cue indicated the prior probability of occurrence of each stimulus category ("L": 25% CW, 75% CCW; "N": 50% CW, 50% CCW; "R": 75% CW, 25% CCW). The cue was shown at the start of a trial and remained on the screen until a response had been registered. Participants received categorical feedback about choice accuracy, before continuing to the next trial. Feedback was based on the average orientation of the array shown on the screen. **c** In Experiment 2, after having made a choice, participants estimated the probability that the choice was correct by sliding a marker along a scale (50–100% in increments of 1%). **d** In Experiment 3, participants could opt out of making a choice and receive "correct" feedback with a 75% probability. **e** In Experiment 4, after having made a choice, participants were required to categorise (low versus high) either the contrast or the variability of the stimulus array. Here we show a contrast trial

we applied signal detection theory[28,29] to quantify the degree to which participants shifted their decision criterion in accordance with the prior cue (see Methods). Briefly, we constructed a bias index, which was computed as the difference in the decision criteria between the condition where the prior cue was clockwise and the condition where the prior cue was counter-clockwise. The higher the bias index, the higher the influence of the prior cue on choice.

As expected under an ideal observer framework, participants used the prior cue more in the low-$c$ than in the baseline condition ($t$-test; $t(39) = 4.89$, $p < 0.001$; Fig. 2c). However, contrary to an ideal observer framework, participants used the prior cue less in the high-$v$ than in the baseline condition ($t$-test; $t(39) = 2.85$, $p < 0.01$; Fig. 2c). This pattern is clear from psychometric curves created separately for each condition (compare inflection points in Fig. 2b). Consistent with these results, a full factorial analysis of the bias index identified a negative main effect of contrast (ANOVA; $F(1,39) = 24.5$, $p < 0.001$) and a negative main effect of variability (ANOVA; $F(1.9,75.7) = 10.0$, $p < 0.001$; Fig. 2d). Finally, including both neutral and biased trials, we used trial-by-trial logistic regression to investigate how contrast ($c$) and variability ($v$) affected the influence of the prior cue and sensory evidence ($\mu\theta$) on choices ($\mu\theta$, cue, $\mu\theta^*c$, $\mu\theta^*v$, cue$^*c$, cue$^*v$; Fig. 2e). The prior cue had a larger influence on choices on low-contrast compared to high-contrast trials ($t$-test; $t(39) = 4.05$, $p < 0.001$) and on low-variability compared to high-variability trials ($t$-test; $t(39) = 5.21$,

$p < 0.001$). Taken together, these results show that participants did not adapt to the additional noise arising during integration of variable pieces of information.

**Overconfidence under integration noise.** To test whether participants did not adapt because they were blind to integration noise, we analysed the confidence reports elicited in Experiment 2 (Fig. 1c). We implemented a strictly proper scoring rule such that it was in participants' best interest (i) to make as many accurate choices as possible and (ii) to estimate the probability that a choice is correct as accurately as possible[30]. In support of our hypothesis, analysis of the full factorial design showed that, while mean confidence (we did not analyse other statistical moments) varied with contrast (ANOVA; $F(1,19) = 32.9$, $p < 0.001$), it did not vary with variability (ANOVA; $F(1.1,22.7) = 0.64$, $p > 0.5$). In addition, direct comparison between the low-$c$ and high-$v$ conditions showed that participants were more confident in the high-$v$ condition ($t$-test; $t(19) = 3.97$, $p < 0.001$; Fig. 3a), with participants overestimating their performance (difference between mean confidence and mean accuracy in the high-$v$ condition, $t$-test; $t(19) = 5.60$, $p < 0.001$; greater overconfidence in the high-$v$ than in the low-$c$ condition, $t$-test; $t(19) = 2.70$, $p < 0.05$; Fig. 3c). Although participants reported lower confidence in the high-$v$ condition compared to baseline (Fig. 3a), this decrease was due to participants utilising response times (RTs) as a cue to confidence[31]: a trial-by-trial regression analysis

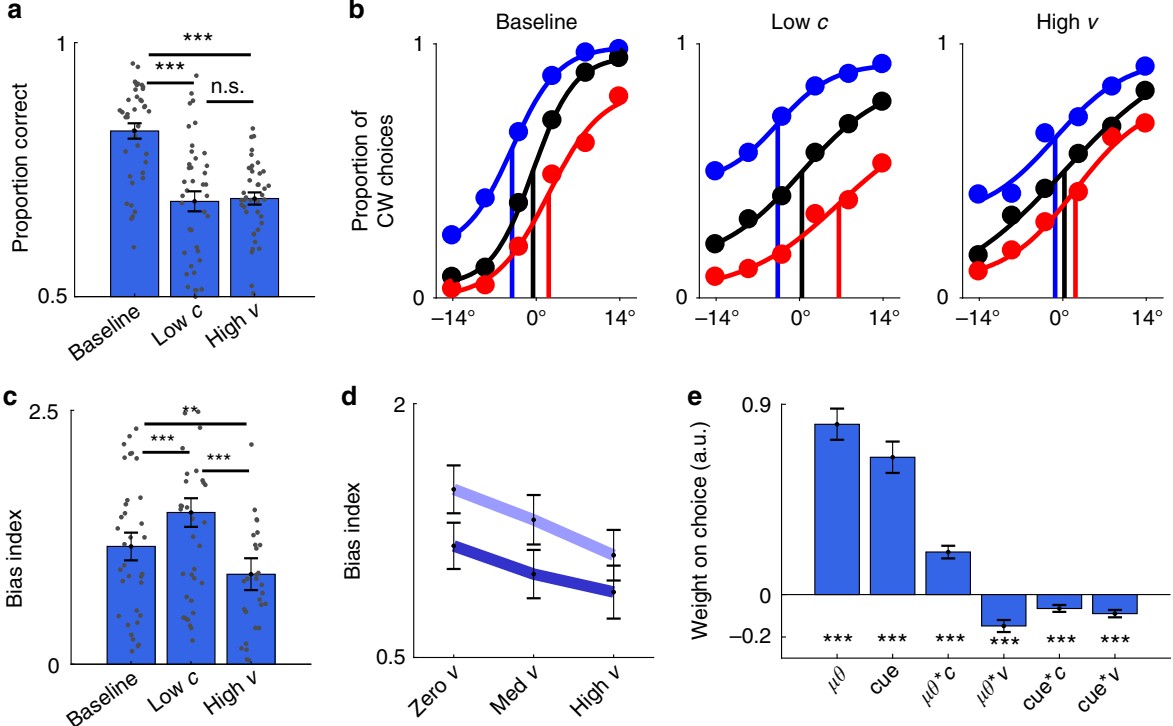

**Fig. 2** Effects of contrast and variability on choice behaviour. **a** Choice accuracy for the baseline, reduced contrast (low-*c*) and increased variability (high-*v*) conditions. **b** Psychometric curves are shallower in the low-*c* and high-*v* conditions compared to baseline. The *x*-axis indicates average orientation relative to horizontal (negative: CCW; positive: CW). Choices shift towards the cued category on biased trials (blue: 75% CW; red: 25% CW) compared to neutral trials (black), but less so in the high-*v* condition. Vertical lines mark the inflection points of psychometric functions fitted to the average data. Psychometric curves were created for illustration. **c** Bias index, a measure of cue usage, is higher in the low-*c* condition, but lower in the high-*v* condition, compared to baseline. **d** Factorial analysis of the effects of contrast and variability on the bias index shows a decrease with contrast (dark blue: high contrast; pale blue: low contrast) and a decrease with variability. **e** Trial-by-trial influence of prior cue on choices measured using logistic regression. $\mu\theta$: signed mean orientation; cue: signed prior cue; *c*: contrast; *v*: variability. **a–e** Data are represented as group mean ± SEM. *$p < 0.05$, **$p < 0.01$, ***$p < 0.001$. For panel **a**, only neutral trials were used. For panels **b** and **e**, both neutral and biased trials were used. For panels **c** and **d**, only biased trials were used

showed that confidence decreased with longer RTs and was unaffected by variability once RTs had been accounted for (*t*-tests; *v*, $t(19) = 0.38$, $p > 0.7$; other predictors, all *t*-values > 4, all $p < 0.001$; see Fig. 3b, see Supplementary Note 7 and Supplementary Fig. 7 for a discussion between RTs and confidence).

In Experiment 3 ($n = 18$), because explicit confidence reports can be highly idiosyncratic[32–34], we obtained an implicit, but perhaps more direct measure of confidence (Fig. 1d)[35–37]. Specifically, on half of the trials (optional trials), we introduced an additional choice option, an opt-out option, which yielded "correct" feedback with a 75% probability. On the other half of trials (forced trials), participants had to make an orientation judgement. Under this design, to maximise reward, participants should choose the opt-out option whenever they thought they were less than 75% likely to make a correct choice. Despite matched levels of choice accuracy in the low-*c* and the high-*v* conditions (*t*-test; forced trials, $t(17) = 0.26$, $p > 0.7$), participants decided to make an orientation judgement more often on high-*v* than on low-*c* trials (*t*-test; optional trials, $t(17) = 2.35$, $p < 0.05$; Fig. 3d), again indicating overconfidence in the face of integration noise. A full factorial analysis verified that the proportion of such opt-in trials varied with contrast (ANOVA; $F(1,17) = 21.3$, $p < 0.001$) but not with variability (ANOVA; $F(1.4,24.0) = 3.5$, $p > 0.05$). Similarly, a trial-by-trial logistic regression showed that the probability of opting in varied with contrast (*t*-test; $t(17) = 6.93$, $p < 0.001$) but not with variability (*t*-test; $t(17) = 1.6$, $p > 0.1$), after controlling for other task-relevant factors. Overall, participants' confidence (probed explicitly or implicitly) was

lower when encoding noise was high, but not so when integration noise was high, despite making a comparable proportion of errors in the two conditions. These results indicate that participants were blind to integration noise.

**Computational model of noise blindness.** We next compared a set of computational models based on an ideal observer framework to provide a mechanistic explanation of the observed data (see Methods and Supplementary Table 1 for an overview of all models considered). There are broadly three components to our approach. First, a generative (true) model which describes the task structure and how noisy sensory evidence is generated. Second, an agent's internal model of the task structure and how sensory evidence is generated. Critically, the internal model may differ from the generative model. Third, a Bayesian inference process which involves inverting the internal model in order to estimate the probability of a stimulus category given sensory evidence and in turn make a response. This process involves marginalising over contrast and variability levels according to a belief distribution over the different experimental conditions. Optimal behaviour can be said to occur when there is a direct correspondence between the generative model and the agent's internal model. We evaluated the models both qualitatively (i.e. model predictions for critical experimental conditions) and quantitatively (i.e. BIC scores).

We focus here on an omniscient model, which has perfect knowledge of the task structure and how sensory evidence is

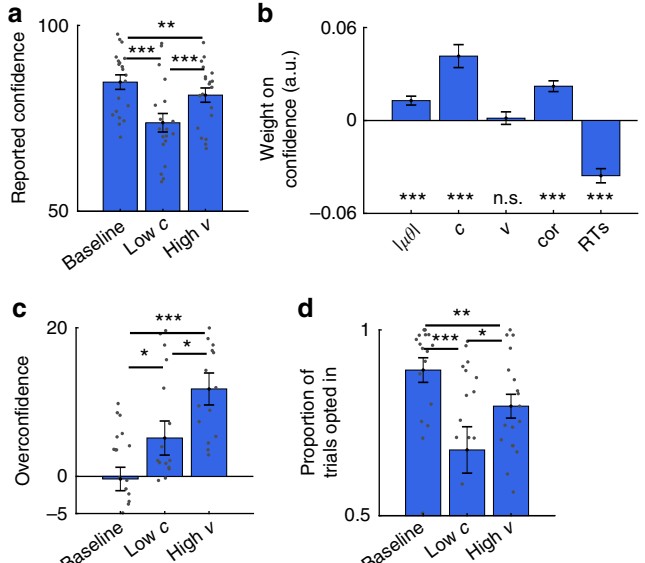

**Fig. 3** Effects of contrast and variability on explicit and implicit markers of confidence. **a** Mean confidence in the baseline, reduced contrast (low-*c*) and increased variability (high-*v*) conditions. **b** Trial-by-trial confidence is not influenced by variability (*v*) but is influenced by the deviation of the average orientation from horizontal ($|\mu\theta|$), contrast (*c*), choice accuracy (cor), which was included to account for error detection[32], and response times (RTs). **c** Overconfidence, the difference between mean confidence and mean choice accuracy, is highest in the high-*v* condition. **d** Higher probability of making a choice (and not opting out) in the high-*v* condition compared to the low-*c* condition. See Supplementary Note 8 and Supplementary Fig. 8 for direct quantification of the accuracy gain foregone by participants in the high-*v* condition. **a**–**d** Data are represented as group mean ± SEM. For all panels, only neutral trials were used

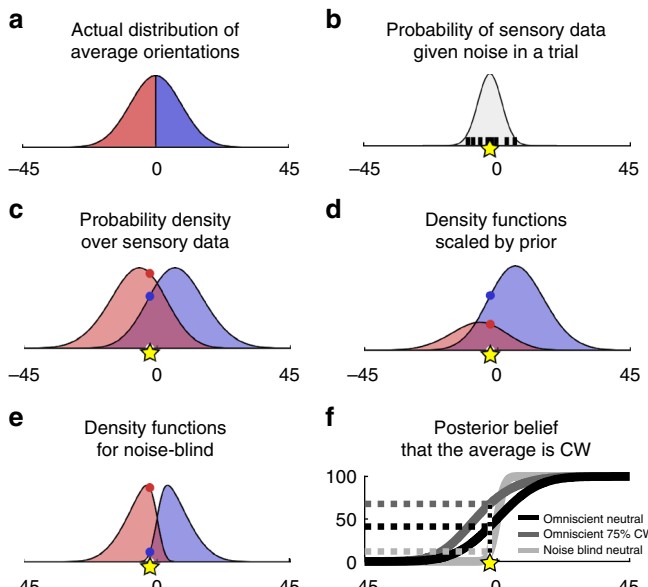

**Fig. 4** Computational model. **a** Distribution of average orientations conditioned on CCW (red) and CW (blue). **b** An agent's sensory evidence was modelled as a sample from a Gaussian distribution centred on the average orientation of the eight gratings in a stimulus array (black vertical lines), with the variance of this distribution determined by both encoding noise and integration noise. The yellow star marks the sensory evidence for an example trial. **c** An omniscient agent has a pair of category-conditioned probability density functions over sensory evidence for each experimental condition (i.e. contrast × variability level; here a single condition is shown). The agent uses the relevant pair of density functions to compute the probability of the observed sensory evidence (yellow star) given each category (red and blue dots). Note that, for an omniscient agent, these density functions match the true probability density over sensory evidence under the generative model. See an example of a full set of density functions for an experiment in Supplementary Fig. 1. **d** Density functions from panel c after scaling by the prior cue (here 75% CW). The sensory evidence (yellow star) is now more likely to have come from a CW stimulus than a CCW stimulus. **e** The noise-blind model only takes into account encoding noise: the density functions therefore overlap less than in panel **c** and they do not match the true probability density over sensory evidence. **f** Posterior belief that the stimulus is CW as a function of the same sensory evidence (yellow star) for the examples shown in panels **c** (black, omniscient model on neutral trials), **d** (dark grey, omniscient model when prior cue is 75% CW) and **e** (light grey, noise-blind model on neutral trials). Steeper curves indicate higher confidence; choice accuracy (on neutral trials) is the same for all models. The variability-mixer curve would have intermediate slope between that of the omniscient and the noise-blind model in conditions of high variability

generated, and two suboptimal models, which provide different accounts of participants' lack of sensitivity to the performance cost associated with stimulus variability. The suboptimal models relax the omniscient assumptions about an agent's beliefs about (i) the task structure and/or (ii) the sources of noise in play.

In our task, the distribution of average orientations was common across experimental conditions and consequently independent of contrast and variability (Fig. 4a). We therefore modelled an agent's sensory evidence as a random (noisy) sample from a Gaussian distribution centred on the average orientation of the stimulus array (Fig. 4b), with the variance of this distribution determined by both encoding noise and integration noise. We verified that the results reported below are not due to this simplifying assumption (see Supplementary Note 2 and Supplementary Fig. 2 where we simulate data under a Bayesian model which operates with eight noisy samples, one for the orientation of each grating, rather than one noisy sample). We used each participant's data from the neutral trials to parameterise their levels of encoding noise and integration noise in each experimental condition (see Methods). The fitted noise levels, which are part of the generative model, were the same for all models; no additional free parameters were fitted to the data and the models only differed with respect to their assumptions about the internal model.

The omniscient model has, for each experimental condition, a pair of functions that specify the probability density over sensory evidence given a CW and a CCW stimulus, taking into account both encoding and integration noise. As the model can identify the current condition (e.g., knows with certainty that a trial is drawn from the high-contrast, high-variability condition), it only

uses the relevant pair of density functions to compute the probability of the observed sensory evidence given a CCW and a CW category (Fig. 4c). On neutral trials, each category is equally likely, and the agent computes the probability that a stimulus is CW and CCW directly from the density functions. On biased trials, the categories have different prior probabilities, and the agent scales the density functions by the prior probability of each category as indicated by the prior cue (Fig. 4d). After having calculated the probability that a stimulus is CW and CCW, the agent can compute a choice (i.e. chose the category with the higher posterior probability) and confidence in this choice (i.e. the probability that the choice is correct).

We now consider two alternative explanations of the participants' lack of sensitivity to the performance cost associated with stimulus variability. First, a variability-mixer model which

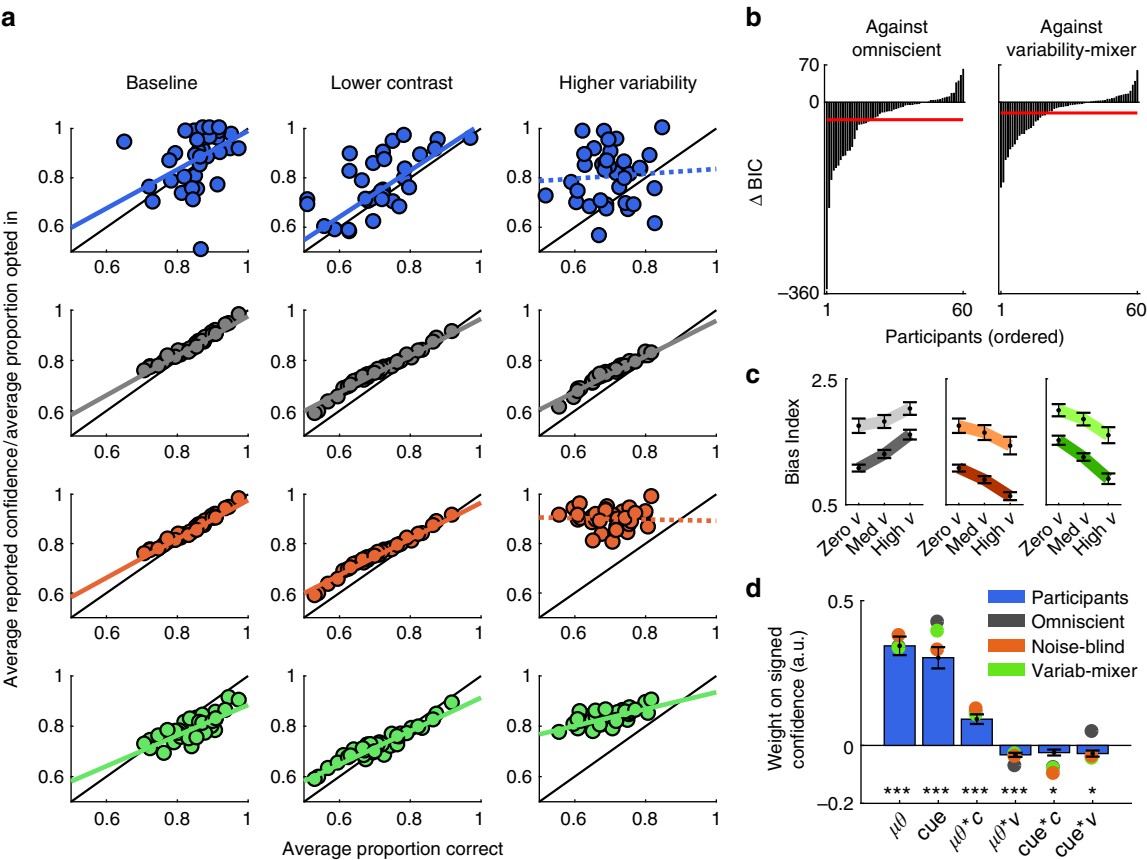

**Fig. 5** Comparison of model and human behaviour. **a** Correspondence between mean accuracy and mean confidence (explicit or implicit) for participants (blue, data from Exp2–3) and the omniscient (grey), noise-blind (orange), and variability-mixer (green) models in the critical experimental conditions. Coloured lines indicate best-fitting slope of a linear regression analysis: solid for $p < 0.01$, dashed for $p > 0.2$. **b** Model comparison (Exp1–3) suggests strong evidence in favour of the noise-blind model over the omniscient model (left panel, average $\Delta$BIC = −32.9) and also over the variability-mixer model (right panel, $\Delta$BIC = −20.4). **c** Omniscient model (left) makes opposite predictions to noise-blind model (middle) and variability-mixer model (right) for the influence of the prior cue on choice (Exp1–2) as variability increases (positive versus negative slopes) but similar predictions as contrast decreases (lighter lines above darker ones). Dark colours: high contrast. Light colours: low contrast. **d** Trial-by-trial analysis of signed confidence (Exp2; negative for CCW and positive for CW) for participants (blue) and the omniscient (grey), noise-blind (orange), and variability-mixer (green) models. **a–d** For panel **a**, only neutral and optional trials were used. For panels **b** and **c**, only biased trials were used. For panel **d**, both neutral and biased trials were used. Within-model variability in predictions comes from variability in encoding and integration noise across participants. For panels **c** and **d**, data are represented as group mean ± SEM

relaxes the assumption that an agent can identify the current variability condition. Specifically, the model uses a single pair of density functions for all variability conditions (i.e. a mixture of density functions across variability conditions). As a result, compared to the omniscient model, the density functions are wider on low-variability trials but narrower on high-variability trials. Second, a noise-blind model which relaxes the assumption that the agent is aware of integration noise. As for the variability-mixer model, the noise-blind model uses a single pair of density functions for all variability conditions, but, critically, these density functions do not reflect the additional noise due to stimulus variability. Because of these differences in the internal model used for Bayesian inference, the three models differ in the degree of confidence in a choice for a given sensory evidence (Fig. 4f) and, by extension, the influence of the prior cue on choice on biased trials.

In support of our hypothesis, the noise-blind model provided the best fit to our data. First, the noise-blind model, and not the omniscient model, predicted three key features of participants' behaviour: (i) overconfidence on high-variability trials within participants (Supplementary Note 3 and Supplementary Fig. 3), (ii) no correlation between mean accuracy and mean confidence

across participants on high-variability trials (Fig. 5a), and (iii) a diminished influence of the prior cue on high-variability trials, as revealed by both the bias index (Fig. 5c) and the trial-by-trial regression predicting (signed) confidence (Fig. 5d), where the prior cue has a positive effect on confidence but less so when variability is high. Second, quantitative comparison yielded "very strong evidence"[38] for the noise-blind model over the omniscient model, with an average $\Delta$BIC across participants of −32.9 (Fig. 5b). Finally, analyses of the patterns of overconfidence in the critical conditions of our factorial design favoured the noise-blind over the variability-mixer model (Supplementary Fig. 3), and quantitative comparison yielded "very strong evidence" for the noise-blind over the variability-mixer model ($\Delta$BIC = −20.4, Fig. 5b). In summary, the modelling indicates that participants neglected integration noise altogether.

**Participants are noise blind and not variability blind.** To further rule out the hypothesis that participants were simply unable to discriminate the variability conditions as proposed by the variability-mixer model, we ran Experiment 4 ($n = 23$). After having made a choice, participants were asked to categorise either

the contrast of the stimulus array (rmc, high: 0.6 vs. low: 0.15) or the variability of the stimulus array (std, high: 10° vs. low: 0°) (Fig. 1e). Again, choice accuracy on neutral trials in the low-$c$ and the high-$v$ conditions was statistically indistinguishable (t-test; $t(22) = 1.22$, $p > 0.2$). We reasoned that, if participants had difficulty identifying the variability condition but were otherwise aware of integration noise, then they should behave closer to optimal when they correctly identified the variability condition. To test this prediction, we used the biased trials to compare cue usage when the variability condition was correctly and incorrectly categorised ($75.71 \pm 2.26\%$ of the variability judgements were correct). In contrast to the prediction, but in line with our hypothesis, participants showed blindness to integration noise even when they correctly identified the variability condition: participants were more biased on low-$c$ than high-$v$ trials regardless of whether the variability categorisation was correct (t-test; $t(22) = 3.03$, $p < 0.01$) or incorrect (t-test; $t(22) = 2.96$, $p < 0.01$; Fig. 6a, b).

In Experiments 1–4, the experimental conditions were interleaved across trials, which may have made it too difficult for participants to separate the different sources of noise in play. To further test the generality of our results, we ran Experiment 5 ($n = 24$) in which either the contrast or the variability level was kept constant across a block of trials (Fig. 6c, d). Even then, and despite receiving trial-by-trial feedback, participants were not more influenced by the prior cue when variability was high compared to the baseline condition (t-test; biased trials, $t(23) = 0.80$, $p > 0.4$), but they were more influenced by the prior cue when contrast was low than when variability was high (t-test; biased trials, $t(23) = 2.37$, $p < 0.05$).

**Sequential sampling account of noise blindness.** A recent study investigated how stimulus volatility (i.e. changes in evidence intensity within a trial) affected choice and confidence[39]. Participants were found to make faster responses and report higher confidence when stimulus volatility was high. These results were explained by a sequential sampling model which assumes that observers are unaware of stimulus volatility and therefore, unlike an omniscient model, adopt a common choice threshold across trial types. We show, using empirical and computational analyses, that this model cannot explain our results (Supplementary Note 4 and Supplementary Fig. 4). For example, the model predicts faster RTs on high-variability than low-variability trials, a prediction which is at odds with our observation of slower RTs on high-variability trials. To further evaluate how our experimental manipulations affected the choice process, we fitted a hierarchical instantiation of the drift-diffusion model[40] to participants' choice behaviour on neutral trials. In line with the above results, this analysis showed that the effects of contrast and variability on accuracy and RTs were captured by a change in drift-rate and not in threshold or non-decision time (Supplementary Note 5 and Supplementary Fig. 5).

**Noise blindness cannot be explained by subsampling.** We have proposed that stimulus variability impairs performance because of noise inherent to cognitive integration of variable pieces of information. An alternative explanation of the performance cost for high stimulus variability is that participants based their responses on a subset of gratings rather than the full array. Under this subsampling account, choice accuracy for high-variability stimuli is lower because of a larger mismatch between the average orientation of the full array and the average orientation of the sampled subset. Here we provide several lines of evidence against the subsampling account (also Supplementary Note 6 and Supplementary Fig. 6).

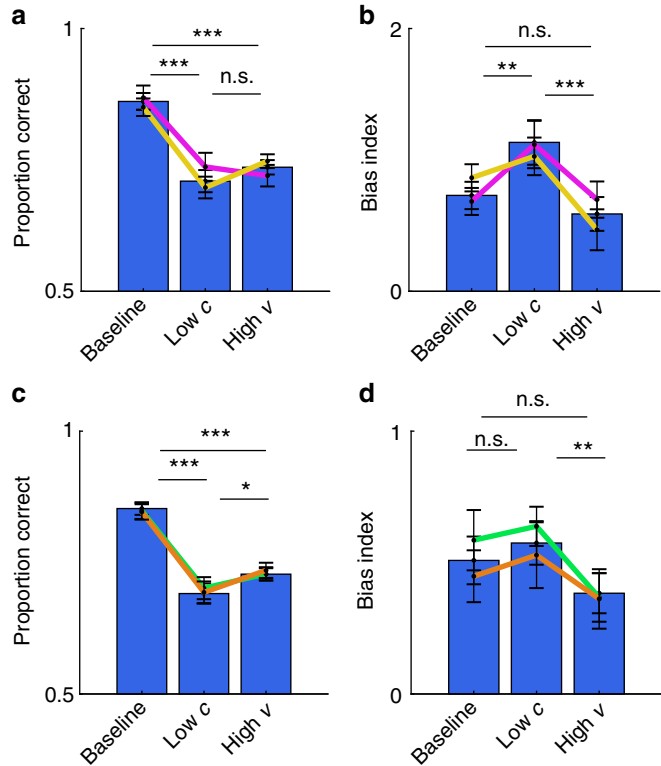

**Fig. 6** Experimental evidence against variability mixing. **a** Choice accuracy for the baseline, reduced contrast (low-$c$), and increased variability (high-$v$) conditions for Experiment 4. **b** In Experiment 4, the influence of the prior cue is highest when encoding noise is high (low-$c$) and lowest when integration noise is high (high-$v$). **c** Same as panel **a**, but for Experiment 5. **d** Same as panel **b**, but for Experiment 5. **a**, **b** Coloured lines indicate trials where the categorisation of contrast was correct (pink) and where the categorisation of variability was correct (yellow). **c**, **d** Coloured lines indicate trials where the contrast level was blocked (green) or when the variability level was blocked (brown). We note that the difference in bias index for the low-$c$ condition between contrast blocking and variability blocking can be explained by a general shift in the bias index according to block difficulty: when contrast is blocked, the low-$c$ condition is accompanied by the hardest condition (the condition with low contrast and high variability), but when variability is blocked, the low-$c$ condition is accompanied by the easiest condition (the condition with high contrast and zero variability). **a**–**d** Data are represented as group mean ± SEM. For panels **a** and **c**, neutral trials were used. For panels **b** and **d**, biased trials were used

We first examined performance under different set-sizes in Experiment 6 ($n = 20$) where the stimulus array was made up of either four or eight gratings (average orientations and orientation variability were equated across set-sizes). We reasoned that, if participants did indeed engage in subsampling, then performance should be higher for four than eight gratings: sampling four items should impair performance in the high-$v$ condition for an eight-item array but not for a four-item array because the average orientations were matched across experimental conditions. However, in contrast to the subsampling account, we found no effect of set-size on choice accuracy (ANOVA; $F(1,20) = 0.004$, $p > 0.9$; Supplementary Fig. 6a), with the effects of contrast (ANOVA; $F(1,20) = 39.6$, $p < 0.001$) and variability (ANOVA; $F(1,20) = 30.3$, $p < 0.001$) comparable to those observed in the previous experiments.

We next simulated performance for eight-grating arrays under a subsampling agent which did not have integration noise but instead sampled a subset of the items (1–8 items, Supplementary

Fig. 6b). The observed difference in participants' performance between the baseline and the high-$v$ conditions could be explained by assuming an agent that sampled about four items out of eight. However, this account—because there is no integration noise—predicts that participants should have similar levels of performance for the baseline and the high-$v$ conditions for four-item arrays, a prediction which is at odds with our data (Supplementary Fig. 6a). If integration noise is introduced, then most, if not all, items would have to be sampled to account for the data.

Finally, we fitted a computational model to participants' choices in Experiments 1 to 3 (eight-item arrays) in order to directly estimate the number of items sampled by each participant. This modelling approach revealed that the majority of participants (42 out of 60) sampled all eight items (Supplementary Table 2). We emphasise that subsampling, even if an auxiliary cause of integration noise, cannot by itself explain participants' lack of sensitivity to the performance cost associated with high-variability stimuli.

## Discussion

Here we propose a new explanation for the previously reported gap in optimality between perceptual and cognitive decisions. Using a novel paradigm, we show, within a single task, that humans are sensitive to noise present during sensory encoding, in keeping with previous perceptual studies[1,8], but blind to noise arising when having to integrate variable or discordant pieces of information, often a requirement in cognitive tasks. This noise blindness gave rise to two common signatures of suboptimality found in cognitive studies: base-rate neglect and overconfidence.

We provided several lines of evidence for our hypothesis. We showed that, when stimulus variability was high, participants were overconfident, as indicated by cue usage, confidence reports, and opt-in responses, even though they received trial-by-trial feedback. We found overconfidence even when stimulus variability was salient (Exp1–3), accurately categorised (Exp4), or constant across a block of trials (Exp5). Overall, these lines of evidence indicate that, while participants were able to track stimulus variability, they simply neglected the performance cost associated with high-variability stimuli. Consistent with this interpretation, the best-fitting computational model of our data indeed assumed that participants were blind to the additional noise inherent to cognitive integration of variable pieces of information.

An extensive literature has considered the different types of noise which affect human choices[21,22,41]. Our classification is partially related to a previous distinction between noise which originates inside the brain, such as intrinsic stochasticity in sensory transduction[42], and noise which arises outside the brain, such as a probabilistic relationship between a cue and a reward[43]. Specifically, our account classifies noise according to when it arises during the information processing that precedes a choice. Encoding noise refers to noise accumulated up to the point at which a stimulus is encoded. As such, encoding noise includes both external noise (e.g., a weak correspondence between a retinal image in dim lighting and the object that caused the image) and internal noise (e.g., intrinsic stochasticity in sensory transduction). By comparison, integration noise strictly refers to internal noise which arises at later stages of information processing, when two or more pieces of information need to be integrated either in space or time, within a limited-capacity cognitive system in order to make a choice. There are several potential contributors to such noise. For instance, errors of inference and information updating[44], information decay in working memory[45], temporal biases such as recency and primacy[46], and conflict among

relevant information[26,27]. Of course, choices may be affected by other types of noise than those considered here. For example, cognitive decisions may involve memories, sometimes distant in the past, and risk and ambiguity[47,48].

Many psychophysical tasks confound encoding and integration noise. For instance, in the classic random dot-motion task, decreasing motion coherence is typically thought to increase encoding noise, as momentary evidence is less indicative of the underlying motion direction[49,50]. However, decreasing motion coherence can also affect integration noise, as momentary evidence becomes more variable relative to a running estimate of motion direction. Indeed, recent work has shown that noisy cognitive inference, related to our notion of integration noise, is a major driver of variability in choices[51]. Similarly, it has been shown that for complex inference problems, a mismatch between an agent's internal model of a task and the true structure of a task provokes departures from optimality[41]. Here we extend these findings by introducing noise blindness as an additional driver of suboptimal cognitive inference. Choice variability due to integration noise, or imperfect inference, may not systematically bias choices away from the correct choice. Blindness to these sources of choice variability, however, predicts systematic overconfidence as reported in the current study. In short, we show that suboptimality can arise not only from having the wrong model of a task but also from having the wrong model of oneself.

We recognize that using our task alone it is hard to categorically say whether the locus of integration noise is early (e.g., in early sensory cortex) or late (e.g., higher association cortex where information is combined over broader windows in space and time). However, several lines of evidence indicate a late locus. First, participants can detect stimulus variability (Exp4), which we would not expect if the sensory representations themselves were distorted during early processing stages, and in support of a higher-order account, noise blindness does not depend on whether variability was accurately discriminated (Fig. 6a, b). Second, the effect of stimulus variability on participants' choice accuracy does not depend on set-size (four vs. eight items in Exp6), and again in support of a higher-order account, neither did the observed noise blindness. Third, participants' RTs and the drift-diffusion modelling suggest that information processing continues after stimulus offset and is thus unlikely to occur in early sensory areas. Fourth, stimulus variability is reflected in brain activity in higher association and control areas (e.g., parietal cortex, anterior insula, and dorsomedial prefrontal cortex)[52], which is consistent with integration noise arising after sensory encoding. More generally, our study is an example of the emerging use of perceptual tasks as a window onto general principles of cognition and decision-making[53].

We do not know why humans are blind to integration noise. One possibility is that basing decision strategies on all sources of noise would prolong deliberation and thus reduce reward rates, or that recognising one's own cognitive deficiencies requires a much longer timeframe. However, a well-known cognitive illusion may help understand why blindness to one's own cognitive deficiencies may not be catastrophic: even though failures to detect salient visual change suggests that cognitive processing is highly limited[54], humans enjoy rich, vivid visual experiences of cluttered natural scenes. Human information processing is sharply limited by capacity, but as agents we may not need to be aware of the extent of these limitations.

## Methods

**Participants**. One hundred and twenty-eight healthy human participants with normal or corrected-to-normal vision were recruited to participate in six experiments (86 females, 10 left-handed, mean age ± SD: 25.00 ± 4.32; Exp1: $n = 20$; Exp2: $n = 20$; Exp3: $n = 20$; Exp4: $n = 23$; Exp5: $n = 24$; Exp6: $n = 21$). Participants

were reimbursed for their time and could earn an additional performance-based bonus (see below). The experiments were conducted in accordance with local ethical guidelines and all participants provided written informed consent. The study was approved by the University of Oxford Central University Research Ethics Committee.

**Experimental paradigm.** All six experiments were based on the same psychophysical task. On each trial, participants had to judge whether the average orientation of a circular array of gratings (Gabor patches; see Fig. 1) was tilted clockwise (CW) or counter-clockwise (CCW) relative to horizontal. The average orientation of the gratings in each trial was randomly selected from a mixture of two Gaussian distributions (centred at 3° either side of the horizontal axis, respectively, and with 8° of standard deviation). We manipulated encoding noise and integration noise by varying two features of the array in a factorial way manner: the root mean square contrast (rmc) of the individual gratings, which affects the difficulty of encoding the stimulus array, and the variability of the orientations of the individual gratings (std), which affects the difficulty of integrating orientations across the stimulus array. The number of contrast and variability conditions varied between experiments: in Experiments 1–3, three contrast levels (rmc = {0, 0.16, 0.6}) and three variability levels (std = {0°, 4°, 10°}); in Experiments 4–6, two contrast levels (rmc = {0.15, 0.6}) and two variability levels (std = {0°, 10°}). The stimulus array was presented for 150 ms and was followed by a 3000 ms choice period. Participants indicated their choice by pressing the right (CW) or the left (CCW) arrow-key on a QWERTY keyboard. They received feedback about choice accuracy, before continuing to the next trial. If no response was registered within the choice period, the word "LATE" appeared at the centre of the screen, and the next trial was started. Experiments 1–3 consisted of 1296 trials, divided into 36 blocks of 36 trials each. Experiments 4–6 consisted of 1200 trials, divided into 32 blocks of 40 trials each.

In Experiments 1 and 2, participants were presented with a cue to the prior probability of each stimulus category. The cue was presented 700 ms before the onset of the stimulus array and remained on the screen until a response was registered. An "N" indicated that the two stimulus categories were equally likely, an "R" indicated a 75% probability of a CW stimulus and an "L" indicated a 75% probability of a CCW stimulus. Half of the blocks contained neutral trials ("N") and the other half contained biased trials ("R" or "L"). The blocks were randomised across an experiment. In Experiment 2, after having made a choice, participants were required to indicate the probability that the choice is correct by moving a sliding marker along a scale (50–100% in increments of 1%). In Experiment 3, on half of the blocks, participants could opt out of making a choice and receive the same reward as for a correct choice with a 75% probability. There was no prior cue. In Experiment 4, after having made a choice, participants had to categorize (high vs. low) either the contrast or the variability of the stimulus array. Participants received trial-by-trial feedback about the categorisation judgement. The judgement types were counterbalanced across trials. In Experiment 5, for each block of trials, we fixed the contrast or the variability level while varying the other feature. In Experiment 6, on half of the blocks, the stimulus array consisted of eight gratings and, on the other half of blocks, the stimulus array consisted of four gratings. Further experimental details are provided in the Supplementary Methods.

**Statistical analyses.** All statistics are reported at the group level. We performed two-way analyses of variance (reported throughout the text as ANOVAs) with participants as a random variable to test the effects of contrast and variability on choice accuracy, RTs, cue usage, confidence (Exp2) and opt-in behaviour (Exp3). We performed most analyses of choice accuracy and confidence using neutral trials; analyses of cue usage were naturally based on biased trials. We used multiple linear regression and multiple logistic regression to isolate the effect of variability on confidence and opt-in responses, respectively. One-sample two-tailed $t$-tests (reported throughout the text as $t$-tests) were applied to estimate (i) the significance of the difference between behavioural measures across conditions being different from zero, and (ii) the significance of the mean distribution of regression coefficients being different from zero. For the analyses in Fig. 5a, seven participants were excluded because of excessive opt-out responses, but result were comparable when including them. All $p$-values lower than 0.001 are reported as "$p < 0.001$", $p$-values $\geq 0.001$ but lower than 0.01 are reported as "$p < 0.01$", $p$-values $\geq 0.01$ but lower than 0.05 are reported as "$p < 0.05$". All $p$-values $\geq 0.05$ are reported as higher than the closest lower decimal (e.g., a $p$-value of 0.175 would be reported as "$p > 0.1$"), with exception of $p$-values between 0.05 and 0.1 which are reported as "$p > 0.05$". The degrees of freedom for the ANOVAs are specified using non-integer values when a Greenhouse–Geisser correction has been used to correct for violations of the sphericity assumption.

**Computational modelling.** We first describe the omniscient model, which takes into account encoding and integration noise and can identify which condition a trial is drawn from (i.e. assigns a probability of 1 to the current condition on a given trial). We then describe the variability-mixer model, which takes into account integration noise but cannot distinguish the variability conditions (i.e. assigns equal probability to all variability conditions on a given trial), and the noise-blind model, which entirely neglects integration noise. For completeness, we ran six additional models which varied an agent's awareness of encoding noise and/or ability to discriminate contrast conditions. We only discuss these models in Supplementary Table 1 as they had no support in the empirical data.

We modelled—regardless of the model—an agent's noisy estimate, $x$, of the true average orientation, $\mu$, as a random sample from a Gaussian distribution with mean $\mu$ and variance $\sigma^2$:

$$x = \epsilon(\mu, \sigma^2), \qquad (1)$$

where $\sigma$ is the agent's total level of noise (encoding plus integration noise) in an experimental condition (see below for noise estimation).

We assumed that an omniscient agent's internal model has, for each condition, a unique pair of category-conditioned probability density functions (PDFs) over sensory evidence which reflect the total level of noise and the true probability distribution over average orientations (see Fig. 4c for an example). As such, an omniscient agent would have six pairs of PDFs in Experiments 1–3 and four pairs of PDFs in Experiments 4–6. An omniscient agent uses the relevant pair of PDFs to compute the probability of the sensory evidence given a CW and a CCW category:

$$\text{PDF}_{\text{cat\&cond}} = p(x|\text{cat}, \text{cond}), \qquad (2)$$

where cat is the category and cond is the condition. We constructed the PDFs by convolving the true probability distribution over average orientations with a zero-centred Gaussian distribution with variance $\sigma^2$ depending on a participant's total noise in a condition. Note that the construction of these PDFs is specific to the model in question (see construction of non-omniscient PDFs below) and is the only source of variation in model predictions about choice and confidence.

We assumed that an agent—regardless of the model—would compute the probability of each category using Bayes' theorem:

$$p(\text{cat}|\text{cue}, x, \text{cond}) = \frac{p(x|\text{cat}, \text{cond}) \cdot p(\text{cat}|\text{cue})}{(p(x|\text{cat}, \text{cond}) \cdot p(\text{cat}) + p(x|\text{cat}_{\text{alt}}, \text{cond}) \cdot p(\text{cat}_{\text{alt}}|\text{cue}))} \qquad (3)$$

where $p(x|\text{cat}, \text{cond})$ is computed using the relevant PDFs and $p(\text{cat})$ is the prior probability of the category in question as indicated by the prior cue. If the category in question is CW, then the alternative category, $\text{cat}_{\text{alt}}$ is CCW, and vice versa. On neutral trials, the prior probability of each category is 50%. On biased trials, the prior probability of one category is 75% and the prior probability of the other category is 25%. The computation detailed in Eq. (3) can be thought of as scaling the relevant PDFs by the prior probability of the respective category (see Fig. 4d for an example).

Finally, we assumed that an agent—regardless of the model—makes a decision, $d$, by selecting the category with higher posterior support and computes confidence in this decision as:

$$\text{Confidence} = p(d = \text{cat}|\text{cue}, x, \text{cond}) \qquad (4)$$

which in our task is directly given by the posterior probability of the chosen category.

Because the omniscient model takes into account encoding and integration noise and knows which experimental condition a trial is drawn from, an agent will (i) be appropriately influenced by the prior cue, (ii) accurately estimate the probability of having made a correct choice, and (iii) opt out of trials when being less than 75% likely to be correct. We now describe the two models which relaxed the omniscient assumptions.

We first consider the variability-mixer model which takes into account integration noise but cannot distinguish the different variability conditions. Therefore, when estimating the probability of the sensory evidence given a CW and a CCW category, the variability-mixer marginalizes its estimate over all possible variability conditions (equivalent to an omniscient agent whose PDFs have been mixed across variability conditions). As a result, when orientation variability is low, the PDFs are more overlapping than for the omniscient model. Conversely, when orientation variability is high, the PDFs are less overlapping than for the omniscient model. For these reasons, a variability-mixer model would display a mixture of under- and overconfidence.

Finally, we consider the noise-blind model which neglects integration noise. Like in the case of the variability-mixer model, a noise-blind agent only has a pair of PDFs for each contrast level but, unlike in the case of a variability-mixer model, these PDFs only take into account encoding noise. As a result, when orientation variability is non-zero, the PDFs are less overlapping than under either of the two other models (Fig. 4e). A noise-blind agent would therefore tend to hold stronger posterior beliefs (i.e. steeper curves for Fig. 4f). Such stronger posterior beliefs will lead a noise-blind agent to (i) be less influenced by the prior cue than needed, (ii) overestimate the probability of having made a correct choice, and (iii) not opt out of trials when being less than 75% likely to be correct.

We note that these models (as well as the six additional models described in Supplementary Table 1) make the same predictions about choice on neutral trials but differ with respect to (i) choices on biased trials and (ii) confidence and opt-in behaviour on both neutral and biased trials. Choice probabilities were computed by marginalizing over the distribution of noisy sensory samples on a given trial; note that, for a given sensory sample, the stimulus category with the higher posterior probability is always chosen.

In the aforementioned models, we assumed that an observer's inferences are conditional on the average orientation of a stimulus array. We made this simplifying assumption because, by design, the average orientation is independent of variability in the orientation of individual items across experimental conditions. However, it is possible that, if we modelled an observer's inferences as conditional on the orientation of individual items, then integration noise may not be needed to

account for the performance cost associated with high-variability stimuli. We simulated performance under this ensemble model, and show that it cannot predict the performance cost associated with high-variability stimuli (Supplementary Note 2 and Supplementary Fig. 2).

**Noise estimation.** We assumed that each experimental condition was affected by Gaussian noise with a specific standard deviation, $\sigma_{cond}$. We assumed that encoding noise depends upon the contrast of the array and that integration noise is proportional to the variability of orientations in the array. We estimated the total level of noise for each condition using four free parameters (three for Experiments 4–6). Two parameters characterised the level of encoding noise for each contrast level: one for low contrast ($nC_{low}$) and one for high contrast ($nC_{high}$). The other two parameters (one for Experiments 4–6) characterised the level of integration noise for each variability level: one for medium variability ($nV_{med}$, only for Experiments 1–3) and one for high variability ($nV_{high}$). For a given condition, the total level of noise (the standard deviation of the Gaussian noise distribution), $\sigma_{cond}$, is thus given by:

$$\sigma_{cond} = \sqrt{\left(\varepsilon\sigma_{cond}^2\right) + \left(\iota\sigma_{cond}^2\right)}, \qquad (5)$$

where $\varepsilon\sigma_{cond}$ and $\iota\sigma_{cond}$ specify the contribution of encoding noise and integration noise, respectively. For instance, for the low-contrast, high-variability condition, the total level of noise would be given by substituting $nC_{low}$ for $\varepsilon\sigma_{cond}$ and $nV_{high}$ for $\iota\sigma_{cond}$.

We fitted the four noise estimators for each participant by maximizing the likelihood of the participant's choice using neutral trials only (we used a genetic algorithm with a population size of 100 individuals and a maximum generation time of 1000 generations). We note that, because of our factorial design, we could separate the two sources of noise. We used the fitted parameters for each participant to construct the model PDFs described above. We note that the model predictions pertain to independent features of the data: (i) confidence on neutral trial choices, (ii) choices (and choice probabilities) on biased trials, and (iii) probability of opting out.

The mean ± SEM of the best-fitting values for the four noise parameters ($nC_{low}$, $nC_{high}$, $nV_{med}$ and $nV_{high}$) in units of degrees were: 10.10 ± 1.51, 3.31 ± 0.39, 3.0 ± 0.78 and 6.8 ± 1.0, respectively. Following Eq. (5), the estimated total amounts of noise fitted for the three key conditions (baseline, low-$c$ and high-$v$) were therefore: 3.31 ± 0.39, 10.1 ± 1.51 and 8.0 ± 1.0, respectively. There was a significant difference between the values for the baseline condition and those for the other two conditions (both $p$-values < 0.001), but no significant difference between the low-$c$ and high-$v$ conditions ($p$-value > 0.16).

**Psychometric fits.** We fitted psychometric curves to the average proportion of clockwise choices using a four-parameter logistic function:

$$P = \frac{A_1 - A_2}{1 + e^{(x-x_0)/dx}} + A_2, \qquad (6)$$

where $P$ is the proportion of CW choices, $A_1$ is the right asymptote, $A_2$ is the left asymptote, $x_0$ is the inflection point and $1/dx$ is the steepness, and $x$ is the average stimulus orientation at which the proportion of CW choices is evaluated. We computed the proportion of clockwise choices within six bins (quantiles) over average orientations relative to horizontal. The psychometric curves shown in Fig. 2b are for illustration only.

**Bias index.** We used signal detection theory[28,29] to calculate the decision criteria, $c$, separately for trials on which the prior cue favoured CW and trials on which the prior favoured CCW. The decision criterion provides a signed estimate of the degree to which the prior cue biases a participants' choices independently of their sensitivity to average orientation. We computed the criterion as, $c = -0.5[\Phi^{-1}(HR) + \Phi^{-1}(FAR)]$, where $\Phi^{-1}$ represents the inverse of the normal cumulative density function, and HR and FAR represent the hit rate (i.e. the proportion of CW trials where participants responded CW) and false alarm rate (i.e. the proportion of CCW trials where participants responded CW), respectively. We then used the difference between $c$ when cued CW ($c_{CW}$) and $c$ when cued CCW ($c_{CCW}$) as our measure of cue usage: bias index = $c_{CW} - c_{CCW}$. Higher values indicate greater cue usage. We computed a bias index for each participant and each experimental condition.

**Reporting summary.** Further information on experimental design is available in the Nature Research Reporting Summary linked to this article.

## Data availability
Anonymised behavioural data and code supporting our main analyses are available via the Open Science Framework (https://doi.org/10.17605/OSF.IO/QYNJG).

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

## Acknowledgements

This work was supported by a Wellcome 4-year-PhD grant to S.H.C. (0099741/Z/12/Z) and an ERC starter grant to C.S. (281628). The Wellcome Centre for Human Neuroimaging is supported by core funding from Wellcome (203147/Z/16/Z).

## Author contribution

S.H.C., T.E., D.B. and C.S. conceived the study. S.H.C., J.D., D.B. and C.S. designed the experiments. S.H.C. programmed the experiments. S.H.C., J.D. and D.B. performed the experiments. S.H.C. and D.B. analysed the data. S.H.C., D.B. and R.M. developed the models and performed simulations. S.H.C., R.M., D.B. and C.S. interpreted the results. S.H.C. drafted the manuscript. S.H.C., D.B. and C.S. wrote the manuscript. All authors approved the final manuscript.

## Additional information

**Competing interests:** The authors declare no competing interests.

