## [Peer Review File · Nature Communications]

Reviewers' Comments:

Reviewer #1:

Remarks to the Author:

The manuscript presents experimental results addressing perceptual decision making in the face of different sources of variability. Subjects are asked to report on the average visual orientation of a spatial array of oriented Gabor patches. The authors find that discrimination performance decreases when either the image contrast is lowered (low-c), or the orientation variability across different Gabors is increased (high-v). However, they report that in the high-v condition, but not low-c, subjects are 'blind' to their own reduced performance: compared to a baseline condition, subjects are more confident of their choices even if they are more often wrong, they often ignore cues about the prior probability of the correct answer, and fail to opt out of a wrong choice when given the chance to do so. The manuscript is written clearly, and the analysis and results seem technically sound.

This result is in itself interesting, but I fail to see its generality and the manuscript seems limited in scope. The authors main claim is that their results dissociate perceptual and cognitive sources of variability. But it is not clear why averaging the orientation of the Gabors on screen would be representative of 'cognitive integration' (as opposed to, e.g., a purely perceptual process involving pooling of local visual features). In addition, the notion that there are different sources of variability in perceptual decision making is not new, and I am not sure what this manuscript adds to the literature. In particular, the relation to Drugowitsch et al 2016 deserves much more in depth discussion, as that paper uses a very similar task, and identifies a more intermediate source of variability (imperfect inference), while also providing a much more detailed quantification of all the noise sources.

One aspect of the results that in my opinion deserves more space is the effect on reaction times (RT), as these are sometimes used as a proxy for decision confidence. If I understand correctly the results, RTs are slower in the high-v compared to baseline condition. First, I may have missed it, but it would be interesting to compare RTs in high-v vs. low-c. Second, if one takes RTs as a measure of confidence, it seems to be consistent with subjects being less confident in high-v than baseline. One could argue that RT is a more direct measure of confidence than the numerical scores and the prior cue (which has a rather different format from the actual stimuli). Conversely, if RTs in this experiment are not a good measure of confidence, the authors could explain why so, and also why RTs are indeed longer in high-v vs baseline (and, again, how about low-c?).

The choice of image contrast to manipulate 'sensory' uncertainty should be justified and validated, even though it is an intuitive one. Specifically, there is literature arguing that orientation discrimination thresholds do not change with contrast, at least within the range of contrasts above detection threshold. The values used here of 15% and 60% are substantially above detection threshold. Mareschal and Shapley (Vision Research 2004) argue that this invariance does not hold out for very small stimuli, but in this manuscript the Gabor size is ~ 1 degree which in many of the experiments of Mareschal and Shapley lead to no effect of contrast on discrimination thresholds.

The clarity of Figure 4 could be improved. Panels could use titles/labels. Why not show also the variability-blind model? Panel F needs legend for the different shades and lines.

Section title on line 294 seems incomplete.

Reviewer #2:

Remarks to the Author:

The authors attempt to unify the frequently observed near-optimality for perceptual tasks, and sub-optimality for cognitive tasks into a common framework. They suggest that human subjects are sensitive to low-level sensory noise, but insensitive to high-level variability, resulting in these sub-optimality. The way they show this is through a decision-making task in which they can modulate both qualities. They then show that high-level variability does not correctly impact prior/likelihood weighting, confidence ratings, and opt-out decisions. Based on this they claim that cognitive sub-optimality result from ignorance of this high-level variability, which leads to (in their case) "integration noise".

While the work is timely and important, I am not sure that the work presented in the manuscript in its current state supports the author's conclusions. The two main issues are the possibility of stimulus subsampling, and their models.

Regarding stimulus subsampling, Fig. S4 seems to suggest that there is very little "integration noise" for the 4-item stimulus. If I understand this figure correctly, the 'high v' blue bar in Fig. S4A is the observed performance for 4 items, whereas the red line in Fig. S4B is the performance predicted from a model that assumes only "estimation noise". Comparing these two performance levels would suggest that the subject's performance really well matches the "estimation noise"-only prediction for 4 items. Furthermore, there is no performance boost when moving from 4 to 8 items (Fig. S4A). Thus, what is the evidence that "integration noise" is a real thing, and doesn't only arise from subsampling of around 4 items when 8 items are present?

Regarding modeling, what is currently called the optimal model implicitly assumes knowledge of the stimulus condition before stimulus onset, which is impossible in experiments that interleave different contrasts and variability levels. In these cases, subjects need to simultaneously infer variability level and contrast, as well as the average orientation. That subjects can infer the former two variables is demonstrated in Exp. 4. Optimal inference then proceeds by computing the joint posterior over all three quantities and marginalizing over different variability levels and contrasts, to reach a posterior over average orientation alone that guides the choices. The "variability-blind" model would lack inference of this variability, but instead would operate with the average. The "noise-blind model" would assume no variability whatsoever. Nonetheless, the marginalization is essential for the optimal model and might lead to different optimal model predictions than the ones currently presented in the paper. While they might not explain the data better than the model that is currently in the manuscript, this remains to be shown. The marginalization will certainly introduce biases in the confidence judgments, causing underconfidence for easy conditions and overconfidence for hard conditions, even for the optimal model.

In terms of model fits, could you provide the fitted parameters for the different models?

Additional details:

I32, "choices are optimal": most manuscript suggest near-optimality, not optimality. This also applies to later mentions of "optimal".

I55: "early sources of noise" - unclear if this also refers to noise outside of the decision-maker that can additionally corrupt the decisions (e.g., random-dot motion task).

I137: "measure the degree to which participants shifted their decision criterion" - how exactly is this measured? The methods are not much more informative about this. As there are multiple ways to do this, it should at least be described in Methods in detail.

Fig 2B: Are the fits to the psychometric curve per-subject and averaged, or direct fits of the group average? Can you add "degrees" labels to the horizontal axes?

Fig 3B: How can choice accuracy drive confidence? Choice accuracy is not known to the subject before confidence and thus cannot directly determine confidence. Is it supposed to be a proxy for some unobserved latent variable in the subject's head?

Furthermore, how do prior cues impact confidence? Do they change confidence ratings in the expected way?

Fig 5: shows only the model comparison between two models. What happened to the third model that was fitted to the subjects' data?

Fig 6D: why does blocking results in a higher bias index for the low-c stimuli?

I331: "and therefore, unlike an optimal model, adopt a common choice threshold across trials" - this model is also optimal, but for different underlying assumptions. In fact, those assumptions are more realistic even for the experiments that are described in the manuscript (see above).

I345: "Under the subsampling account, ..." - I don't understand this argument. Isn't the variability of individual orientations the same, independent of if there are 4 or 8 on the screen? If so, wouldn't subsampling a single orientation yield the same amount of variability, independent of the number of items on the screen? The same argument would then apply to subsampling up to 4 orientations - the overall variability should be invariant to the number of orientations on the screen.

Fig S1: "Four pairs of PDFs, one for each of the four conditions in Exp. 3" - weren't there 6 conditions in Exp. 3? 2 contrasts, 3 variability levels?

Fig 3C-E: what is the horizontal axis? Is it the average orientation away from zero? If yes, could you provide the degrees?

Reviewer #3:

Remarks to the Author:

The manuscript addresses a central question in the domain of decision-making: the gap between the apparent optimality of perceptual decisions and lack of it in cognitive ones. This is definitely one of the most challenging puzzles facing this field, and any progress towards solving it is important and of high impact for the field. To address the question, the authors have carried out a comprehensive psychophysical and computational study (6 experiments), which manipulate what they refer to as encoding and integration noise, and examined the impact on two dependent variables: i) the reliance on priors (i.e., biasing due to cues), ii) the degree of confidence in the choice. Using an ideal-observer modeling approach, the authors show that while the observers appear to modulate their use of priors and their confidence (as they should) based on encoding-noise, they appear to not do this for the integration-noise. The authors also distinguish between two sub-models that could account for the neglect of the integration-noise and they provide evidence to support the idea that although the observers can discriminate trials based on integration-noise, they still appear to not take this type of noise into account in their decision-process.

I find the study novel and original, very well executed and the paper clearly written. As the question is also of prime importance to the field, I do believe that a revised manuscript, which should clarify few conceptual issues (helping to further strengthen the paper) and a number of more technical ones

(allowing the readers to better understand it), would provide an excellent contribution to the journal. Below I list a number of such issues, which the authors should try to address in their revision.

Conceptual

1. What about-RT?

The data convincingly shows that the amount of biasing and the degree of confidence are modulated by encoding but not by integration-noise (variability of the orientation). However, what about decision-RT? In the Supplement, it is reported that, in fact, the RT is affected by the variability: observers are slower in deciding about stimuli with high variability. The authors use this to argue against a model that predicted faster-RT for high variability trials. This is fine, but the issue is how does the present model account for the slow down. What about the possibility that, in fact, the observers increase the decision-criterion to compensate for the difficulty of the high-variability condition, accounting thus for the slowdown? If that was the case, we would have a situation, in which, while the observers fail to modulate the priors and confidence, they do adjust the decision criterion. This would indicate a more subtle situation than the one (noise-blindness), which the authors support. Finally, one may also argue, that the observers are somewhat rational in not down-modulating their confidence with variability, if they already tried to compensate for this by taking more samples to improve their choice-performance. The latter would be also consistent with the fact that confidence is still modulated, by RT (RT could be an estimate of the criterion), and the question that may be important to address is whether the benefit from the change of criterion is not efficient enough and why. It is possible that there are other answers to these issues or that I missed something important here. However, I believe that clarifying these issues would make the paper much stronger.

2. Types of noise.

Across the manuscript, there is some lack of clarity about the nature of the two types of noise. In the introduction (lines 55-64), encoding noise is associated with low visibility stimuli, while integration-noise is associated "with multiple, some-times discordant pieces of information". So what about a moving dot stimuli array, in which the coherence of the motion varies in time? (should the noise here be mapped to encoding or to integration?). The authors mention in the same sentence (line 63) the possible limitations due to a capacity limited system (this also comes in the Discussion, lines 381-382). Is this the important distinction? But this would seem to go against the result that the results of Exp-6, which indicate that the capacity limitations are not playing much role in the task (?).

In general, I feel that the paper would benefit from a more clear discussion of the nature of types of noise that the observers have access to and use, and the ones they do not. While this is clear enough for the present study, it is not so for how we should think of generalizing the results to other situations.

A related issue that may also be important to clarify, in this context: I assume that the average orientation in the high-v condition is constrained to be at a fixed value (same as for low-v trials) and this is why the authors refer to this integration-noise as internal (lines 66-67). Perhaps this detail appears in the Suppl, but it is an important one, which should be mentioned in the main text.

Minor & clarifications (in order of appearance in the manuscript)

Fig. 1E. Last panel on the right: this appears the same as the previous one (to its left); should this not refer to VARIABILITY instead of contrast?

Line-139: could be helpful to clarify that the "bias-index" corresponds to CCW-CCCW.

(it took some time to guess this)...

Lines 178-181: Is the confidence affected by contrast, once RT is factored out?

Line 205-6: "probability of opting in did not vary with variability". Fig 3D seems to show a variability effect.

Fig. 4F. The light-gray curve is hardly visible. From what I see, however, this curve is the steepest (thus has the best discriminability; why should the noise-blind model have higher discriminability than the optimal one?). Also the white dots in Fig. 4C-E are quite hard to see.

Lines 262-279. It could help to clarify the differences in the predictions between the two non-optimal models (if possible, illustrate them in a plot). Furthermore, it is not clear enough, how any of the models are used to make predictions about confidence. In particular, it will be important to explain how the models are used to create the Figures in Fig. 5A. What determines the variability we see (in the models)? Perhaps it could also be more efficient to start this paragraph with the Bias-index, as this was discussed more earlier on, before shifting to the confidence, and explaining this in more detail.

Line-323: "but" missing (I think) before "only..."

Lines 337-338. "We have proposed that orientation variability impairs performance because of additional noise inherent to cognitive integration of variable or discordant pieces of information. However, another possible explanation of this relationship is that participants based their choices on a subset of gratings rather than the full array". It is not clear, in fact, why this is ANOTHER explanation (?). If the integration-noise is due to the capacity limitations of WM or of the attentional system, than the second sentence is quite congruent with the first one.

Reviewer #1 (Remarks to the Author):

The manuscript presents experimental results addressing perceptual decision making in the face of different sources of variability. Subjects are asked to report on the average visual orientation of a spatial array of oriented Gabor patches. The authors find that discrimination performance decreases when either the image contrast is lowered (low-c), or the orientation variability across different Gabors is increased (high-v). However, they report that in the high-v condition, but not low-c, subjects are 'blind' to their own reduced performance: compared to a baseline condition, subjects are more confident of their choices even if they are more often wrong, they often ignore cues about the prior probability of the correct answer, and fail to opt out of a wrong choice when given the chance to do so. The manuscript is written clearly, and the analysis and results seem technically sound.

Thank you for the positive comments, and for a thoughtful review of our paper.

This result is in itself interesting, but I fail to see its generality and the manuscript seems limited in scope. The authors main claim is that their results dissociate perceptual and cognitive sources of variability. But it is not clear why averaging the orientation of the Gabors on screen would be representative of 'cognitive integration' (as opposed to, e.g., a purely perceptual process involving pooling of local visual features).

The reviewer is right to raise this issue. In the initial submission, we could have clarified the logic of our experiment in more detail and been more careful in our use of terminology. We have revised our manuscript to address these issues, with a focus on the generality of our approach.

Our study was designed to understand a well-known paradox – that in the perceptual decision literature, humans have been observed to optimally integrate priors and likelihoods (e.g. classic studies by Konrad Körding and others), whereas precisely the opposite is the case in the cognitive literature (e.g. the classic demonstrations of base rate neglect by Kahneman and Tversky). The reviewer is right that our task is closer to a “perceptual” task (as used in the former literature) than a “cognitive” task (as used in the latter). At first glance, it might appear that this limits the scope of our findings to the perceptual domain. However, our task was carefully designed to allow us to manipulate two sources of noise that place strong limitations on performance in (traditional) perceptual and cognitive tasks, namely (i) the cost of encoding any given item, and (ii) the cost of combining multiple potentially conflicting items. In other words, what the reviewer refers to as the “limited scope” – our use of a single task to address this broad question – was in fact precisely what we see as the strength of our experiment: an orthogonal manipulation of the potential costs in previous “perceptual” (encoding information) and “cognitive” (combining information) tasks within a single, well-controlled sensorimotor paradigm. This allowed us to test our hypothesis that differential sensitivity to “encoding noise” and “integration noise” is a major driver of the optimality gap, in contrast to previous accounts that have focussed on differences in the domain, incentive structure and format of the stimulus materials between perceptual and cognitive tasks. We have now revised the introduction (**lines 66-77**) and the discussion (**lines 404, 431-433 and 437-440**) to clarify the logic and generality of our experimental approach.

In the initial submission, we may have been inexact in our terminology in places. We did not mean to imply that our paradigm is a “cognitive” task – rather that our task manipulates one source of loss (namely, the challenge of integrating conflicting information) that is often a key driver of errors in previous cognitive tasks (for example where participants trade off conflicting risk and reward in behavioural economics experiments). In the revised submission, we have modified our terminology to make this clear.

Turning to the reviewer's second point, we acknowledge that the field lacks a satisfying account of why it is hard to combine more variable features. The reviewer is right to point out that spatial averaging judgments may involve pooling of local visual features (e.g. in V4, Freeman & Simoncelli 2011 Nature Neuroscience). In the revised manuscript, we have made sure that this work is appropriately cited. However, we also know that similar costs of feature variability are observed irrespective of whether gratings are averaged in space or in time, and that spatial pooling models would struggle to account for the latter findings (e.g. in the

Drugowitsch study to which the reviewer refers below). Moreover, we have previously used neuroimaging to examine BOLD responses whilst participants make such spatial averaging judgments and found (perhaps surprisingly) that parietal cortex, rather than visual cortex, is sensitive to increases in feature variability (Michael et al 2015, Cerebral Cortex). Together, these findings point to a “late” locus for the cost of integration, beyond perceptual pooling in visual cortex, and whilst there may be multiple potential factors that come into play when humans are asked to average variable information, at least some of these factors might reasonably be termed “cognitive”. We use the term “integration noise” as a generic term for these sources of variability; defining their precise neural and computational basis is an ongoing project in our lab and elsewhere.

More generally, we hope the reviewer will agree that irrespective of any terminological issues, our findings still demand explanation. We find that participants respond very differently to noise incurred by lowering contrast and heightening feature variability. This finding is hard to explain under accounts that assume a common source of perceptual loss for our contrast and feature variance manipulations. It can be explained, however, under the assumption that these two manipulations act at qualitatively different processing stages, and that humans have differential sensitivity to their respective costs.

In addition, the notion that there are different sources of variability in perceptual decision making is not new, and I am not sure what this manuscript adds to the literature. In particular, the relation to Drugowitsch et al 2016 deserves much more in depth discussion, as that paper uses a very similar task, and identifies a more intermediate source of variability (imperfect inference), while also providing a much more detailed quantification of all the noise sources.

We agree with the reviewer that the quantification of different sources of noise in perceptual decision-making is not a new project. However, such a quantification was not our goal. As we detail above, our goal was to test whether, when performing a perceptual task in the presence of integration noise, participants show something akin to base rate neglect (i.e. a suboptimal integration of priors and likelihoods). We are familiar with the important work of Drugowitsch and colleagues (the co-first author of that study is a former lab member) but would argue that the two studies make very different contributions to the literature. First, in their paper they sought to measure the extent to which different sources of noise (sensory, inference, response) corrupt perceptual decisions. By contrast, we sought to measure participants’ sensitivity to such sources of noise. To this end, we deliberately calibrated our task so that encoding noise (contrast) and integration noise (orientation variability) would have comparable effects on performance in neutral conditions (equal priors for both categories), which they did in our experiments. Second, we tested the degree to which participants utilized prior cues, and whether their confidence reflected all or only a subset of the sources of noise. Instead, the Drugowitsch study did not manipulate prior information or measure confidence. Thus, we see the two studies as complementary.

As a side note, the Drugowitsch study shows that “imperfect inference” is the main source of variability in perceptual decisions. The authors of that study agree with us that what we call “integration noise” is a major contributor to imperfect inference – indeed, in follow-up analyses of their dataset, it can be shown that variability in the grating stream is one source of the cost which cannot be accounted for by sensory or motor factors (Valentin Wyart, personal communication).

However, the reviewer is right that we could have discussed these issues in more detail in the main text. In a revised submission, we have now done so (**lines 440-451**).

One aspect of the results that in my opinion deserves more space is the effect on reaction times (RT), as these are sometimes used as a proxy for decision confidence. If I understand correctly the results, RTs are slower in the high-v compared to baseline condition. First, I may have missed it, but it would be interesting to compare RTs in high-v vs. low-c. Second, if one takes RTs as a measure of confidence, it seems to be consistent with subjects being less confident in high-v than baseline. One could argue that RT is a more direct measure of confidence than the numerical scores and the prior cue (which has a rather different format from the actual stimuli). Conversely, if RTs in this experiment are not a good

measure of confidence, the authors could explain why so, and also why RTs are indeed longer in high-v vs baseline (and, again, how about low-c?).

With respect to the first point, the reviewer is right to point this out – in the initial submission, our discussion of the RT analysis is rather scattered throughout the text and may be hard to follow. We have remedied this in the revised submission. We have now grouped these analyses together in the main text, and have added a new section to the Supplementary Information which contains further RT analyses and discussion (section *Response Times*, lines **1026-1060**). This section also includes the plot requested by the reviewer – the average RTs for the three critical conditions (i.e. baseline, low-c and high-v). The plots shows that both the low-c and the high-v conditions have slower RTs than the baseline condition, but that the high-v condition has even slower RTs than the low-c condition. We now discuss these points and the relationship between RT and confidence as requested by the reviewer (lines **1042-1057**).

With respect to the second point, however, we were a little surprised that the reviewer thinks that RTs might be a “more direct” measure of confidence than (1) confidence reports, which were elicited under an incentive-compatible scoring rule (Brier score), and (2) opt-out behaviour, widely used as a proxy for confidence reports in animal studies. While confidence and RT often are partially correlated, we know from the sequential sampling literature that fast responses may reflect high confidence or rapid guesses, whereas slow responses may reflect thoughtful deliberation or great uncertainty (e.g. Pleskac & Bussemeyer Psychological Review 2010). For these reasons, we believe that our approach provides more direct measures of subjective confidence than RTs.

The choice of image contrast to manipulate ‘sensory’ uncertainty should be justified and validated, even though it is an intuitive one. Specifically, there is literature arguing that orientation discrimination thresholds do not change with contrast, at least within the range of contrasts above detection threshold. The values used here of 15% and 60% are substantially above detection threshold. Mareschal and Shapley (Vision Research 2004) argue that this invariance does not hold out for very small stimuli, but in this manuscript the Gabor size is ~1 degree which in many of the experiments of Mareschal and Shapley lead to no effect of contrast on discrimination thresholds.

Thanks for this insightful comment. As the reviewer notes, contrast is an intuitive way to manipulate sensory uncertainty. It is true that some previous studies, such as the Mareschal & Shapley study, have not found an effect of contrast on discrimination thresholds. However, in our study, contrast reduction clearly has a strong effect on discrimination performance (i.e. 10-15% drop in accuracy from high-C to low-C condition). There are several reasons why we might have observed an effect of contrast on discrimination thresholds: for example (i) our stimuli were displayed for a shorter duration (150ms versus 250ms) and (ii) we engineered the noise to have the same spatial frequency properties as the Gabor gratings (building on our previous work, e.g. Wyart et al PNAS 2012). We now make these points in the Methods (lines **781-784**).

The clarity of Figure 4 could be improved. Panels could use titles/labels. Why not show also the variability-blind model? Panel F needs legend for the different shades and lines.

Thanks for these suggestions. We have now added titles/labels as suggested, and added a legend to panel F. We feel that the variability-blind model would overly clutter this already busy figure. The curve for the variability-mixer, would have an intermediate steepness between the omniscient and the noise-blind models, we clarify this in the figure caption (lines **265-267**).

Section title on line 294 seems incomplete.

Thanks, we have corrected the error.

Reviewer #2 (Remarks to the Author):

The authors attempt to unify the frequently observed near-optimality for perceptual tasks, and sub-optimality for cognitive tasks into a common framework. They suggest that human subjects are sensitive to low-level sensory noise, but insensitive to high-level variability, resulting in these sub-optimality. The way they show this is through a decision-making task in which they can modulate both qualities. They then show that high-level variability does not correctly impact prior/likelihood weighting, confidence ratings, and opt-out decisions. Based on this they claim that cognitive suboptimality result from ignorance of this high-level variability, which leads to (in their case) "integration noise".

While the work is timely and important, I am not sure that the work presented in the manuscript in its current state supports the author's conclusions. The two main issues are the possibility of stimulus subsampling, and their models.

We thank the reviewer for recognizing our work as timely and important, and we hope to be able to alleviate his/her concerns below.

Regarding stimulus subsampling, Fig. S4 seems to suggest that there is very little "integration noise" for the 4-item stimulus. If I understand this figure correctly, the "high v" blue bar in Fig. S4A is the observed performance for 4 items, whereas the red line in Fig. S4B is the performance predicted from a model that assumes only "estimation noise". Comparing these two performance levels would suggest that the subject's performance really well matches the "estimation noise"-only prediction for 4 items. Furthermore, there is no performance boost when moving from 4 to 8 items (Fig. S4A). Thus, what is the evidence that "integration noise" is a real thing, and doesn't only arise from subsampling of around 4 items when 8 items are present?

We agree with the reviewer that Fig. S4 in the submitted manuscript might seem to suggest that subsampling provides a good account of our data. However, this is due to a lack of clarity on our part when preparing and presenting the figure. Here we provide arguments as to why Fig. S4 shows evidence *against* subsampling. We start with a short summary of these arguments, and describe them in depth below.

The first argument concerns the equivalent accuracy levels across different set-sizes. Experiment 6 was designed such that participants would show a difference in performance across set-sizes if they engaged in subsampling (i.e. subsampling would be more detrimental to performance for set-size 8 than set-size 4). However, there was no significant effect of set-size on accuracy (Fig. S4A). The second argument concerns the simulations carried out for Fig. S4B which assumes that there is no integration noise. Although the plot shows that the decrease in performance between the baseline and high-v conditions could be explained by sampling only four items from an eight-item array, this account predicts comparable levels of performance in the baseline and high-v conditions for a four-item array, which is not supported by our data. The third argument comes from a new modelling approach where we directly quantify the number of items sampled by participants, showing that for most participants all items must have been integrated. We have carefully revised the manuscript to clarify our arguments and avoid confusion. We have also gathered the arguments against subsampling in a section in the Supplementary Information where we describe the different lines of evidence which support that "integration noise" provides a better account of our data than subsampling (section *Subsampling*, lines 961-1025).

#1 Experimental evidence against subsampling

Experiment 6 was designed to be sensitive to differences in performance across set-sizes if participants were subsampling. The subsampling account predicts that the decrease in accuracy between the baseline condition and the high-v condition is driven by the mismatch between the "true average orientation of all the items in the array" and "the average orientation of the items sampled by participants". The distribution of average orientations was the same for both set-sizes. If participants sampled all items, then there would be no mismatch and thus no difference in performance across the two set-sizes, and any decrease in

performance between the *baseline* and *high-v* conditions would have to have a different explanation. If subsampling did occur, it would have a larger effect on performance on trials with eight-item arrays than on trials with four-item arrays. For instance, if participants could sample ~4 items, there would be NO difference in performance between the *baseline* and the *high-v* conditions for the four-item arrays (because there would be no mismatch), but there would be a performance difference on the eight-item arrays (because half of the items would have been ignored). Experimentally, we found that the decrease in performance between the *baseline* and the *high-v* conditions was consistent across set-sizes and of equivalent magnitude as with all previous experiments (Fig. S4A). This argues against a subsampling account.

The reviewer suggests that a boost in performance should be expected for the eight-item array compared to the four-item array, presumably from averaging out encoding noise. However, the data does not show such a boost: the effect is in a direction consistent with the reviewer's prediction (i.e. accuracy is slightly higher for the eight-item arrays), but it is not significant. One possibility is that there is a trade-off between the number of items which are encoded and the quality with which they are encoded (Van den Berg, Shin, Chou, George, & Ma, PNAS, 2012); such a trade-off may explain why we do not observe a performance boost for the eight-item array.

#2 Simulating performance in the absence of integration noise

These simulations test the hypothesis that the decrease in accuracy between the baseline and high-v conditions is driven by subsampling. The point we wanted to convey with Fig. S4B is that, while sampling four-items could explain the drop in accuracy between the baseline and high-v condition for arrays of eight-items, it cannot explain the drop for arrays of four items. The reviewer is correct that the high-v blue bar in Fig. S4A shows the performance of participants when the stimulus array was composed of four items (which is lower than the baseline blue bar). The reviewer is also correct that the red line (now yellow, to avoid using same colour with different meanings in figure panels) represents the performance of an agent with only estimation noise, showing that about four items would be needed to explain participant behaviour with eight-item arrays. However, if participants indeed sampled four items, then their performance would be matched in the baseline and high-v conditions for four-item arrays, a prediction which was proven wrong by the data. We apologise that our lack of clarity opened the door for misinterpretation.

#3 Quantifying the number of items sampled

To further strengthen our case against subsampling, we fitted a subsampling model to the data of each participant, using three free parameters. The first parameter controls the level of encoding noise in the baseline condition. The second parameter controls the extra amount of noise induced by reducing the contrast of the stimuli. The third parameter controls the number of gratings sampled by the participant. For the majority of participants (42 out of 60), the number of sampled gratings estimated by this model was *eight*. We describe this new modelling approach in the Supplementary Information (section *Subsampling*, lines **1005-1017**).

In summary, we argue that subsampling cannot account for the consistent decrease in accuracy for high-variability conditions – a decrease that is independent of the set-size. Instead, we argue that this pattern is best explained by an increased difficulty for integrating variable stimuli – which we call integration noise. We note that, whatever the source of the decrease in accuracy for high-variability trials, the apparent blindness to this performance drop requires an explanation. Subsampling could have accounted for integration noise, but not for noise blindness.

Regarding modeling, what is currently called the optimal model implicitly assumes knowledge of the stimulus condition before stimulus onset, which is impossible in experiments that interleave different contrasts and variability levels. In these cases, subjects need to simultaneously infer variability level and contrast, as well as the average orientation. That subjects can infer the former two variables is demonstrated in Exp. 4. Optimal inference then proceeds by computing the joint posterior over all three quantities and marginalizing over different variability levels and contrasts, to reach a posterior over average orientation alone that guides the choices.

Thank you for these comments. We recognize that our terminology was potentially confusing and that our description of the modelling was somewhat imprecise. The approach we adopted is in fact similar to what the reviewer requests. Our models do involve marginalization but we failed to state this explicitly. To address the reviewer's comments we have 1) revised the model description and terminology and 2) included additional models to capture the full model space (e.g. marginalization over both variability and contrast or only over variability). We emphasise that the noise-blind model still provides the best account of our data, both in terms of model evidence and model predictions.

First, we have revised the model names. The "optimal" model has been renamed the "omniscient" model to emphasise that the model knows the current condition (see details below). The "variability-blind" model has been renamed the "variability-mixer" model to emphasize that this model, because it does not know from which variability condition each trial is drawn, ends up with mixture of density functions across variability conditions when marginalizing over variability (see details below).

Second, we want to reassure the reviewer that we by no means neglected marginalization. Optimal inference indeed requires that a posterior is computed by marginalizing over the possible contrast and variability levels according to a belief about which condition the current trial is drawn from. However, our models make different assumptions about this step. Models can either have (whether for contrast, for variability or both) 1) perfect knowledge about which condition the trial comes from (i.e. the correct condition has a probability of one and the others a probability of zero), or 2) absolute ignorance (i.e. each condition is assigned an equal probability when performing marginalization). We have revised the description of our modelling, in the main text (**lines 220-321**) as well as in the methods (**lines 536-606**) and in the Supplementary Information (**lines 888-903**) to make these points explicit.

We note that the difference between "noise-blindness" (being unaware of one's integration noise) and "variability-mixing" (mixing integration noise across variability conditions) is clearest when looking at the pattern of overconfidence across experimental conditions. The variability-mixer model shows an easy-hard effect (overconfidence for hard conditions and under-confidence in easy ones), whereas participants and the "noise-blind" model only show overconfidence in conditions of greater integration noise (Fig S2, **lines 904-921**).

The "variability-blind" model would lack inference of this variability, but instead would operate with the average. The "noise-blind model" would assume no variability whatsoever. Nonetheless, the marginalization is essential for the optimal model and might lead to different optimal model predictions than the ones currently presented in the paper. While they might not explain the data better than the model that is currently in the manuscript, this remains to be shown. The marginalization will certainly introduce biases in the confidence judgments, causing underconfidence for easy conditions and overconfidence for hard conditions, even for the optimal model.

We were unclear as to whether the "variability-blind" model described by the reviewer was 1) an interpretation of what we had done, or 2) a suggestion for a new model (which, even if not anchored on an intuition of optimality, could in principle account for the lack of adjustment of behaviour for different variability conditions). Thus, we provide answers to the two possible interpretations. First, we clarify that our variability-mixer model (previously variability blind model) does not operate with the average integration noise but marginalizes over variability conditions, in line with the reviewer's comment on optimality. Second, we decided to test a comprehensive space of nine models which make different assumptions about participants' beliefs and inferences, including three new model variants that operate with the average noise across 1) variability conditions, 2) across contrast conditions, or 3) across both (what the reviewer might have meant by "operate with the average").

A description of the nine models along with quantitative (BIC) comparisons against the best model (noise-blind) is included in the Supplementary Information (**lines 888-903**). Among these is a "full-mixer" model which, in line with the reviewer's suggestion for a more realistic optimal model, does not know which

contrast or variability condition each trial comes from, and thus marginalises (with equal probability for each condition) over both dimensions.

Finally, we thank again the reviewer for prompting us to engage in this more extensive modelling exercise, as this process allowed us to identify a coding bug in the calculation of BIC scores. We wrongly provided prior cues to the simulated agents in Exp3, even though participants did not receive any prior cues. Reassuringly, the quantitative evidence in favour of the noise-blind model (BIC difference) is even stronger after correcting this error.

In terms of model fits, could you provide the fitted parameters for the different models?

Thanks for this suggestion. We realise that this information was not directly clear in the submitted manuscript, but we only fitted the noise parameters to a participant's data (neutral trials) and used the same set of noise parameters for all models to generate model predictions. In other words, the models did not include any free parameters. We now clarify this point in the main text (**lines 240-247**). We have also added information about the best-fitting noise parameters to the Methods where we describe the noise-estimation procedure (**lines 607-637**).

Additional details:

132, "choices are optimal": most manuscript suggest near-optimality, not optimality. This also applies to later mentions of "optimal".

Thanks for pointing this out. We now use "near-optimal" throughout, including in the abstract.

155: "early sources of noise" - unclear if this also refers to noise outside of the decision-maker that can additionally corrupt the decisions (e.g., random-dot motion task).

Yes. We now clarify that "early sources of noise" also refers to noise originating in the outside world (e.g. for our low-contrast grating stimuli [**lines 55-58**]). We have also expanded our discussion of the different sources of noise which may corrupt decisions along these lines (**lines 426,428 and 437-440**).

1137: "measure the degree to which participants shifted their decision criterion" - how exactly is this measured? The methods are not much more informative about this. As there are multiple ways to do this, it should at least be described in Methods in detail.

Our criterion shift index (bias index) is the difference in criterion, c , between the biased trials where the prior cue favoured the clockwise category (C_{CW}) and the biased trials where the cue favoured the counter-clockwise category (C_{CCW}): bias index = $C_{CW} - C_{CCW}$. Criterion c is computed according to the standard signal detection formulation as follows: $c = -0.5[\Phi^{-1}(HR) + \Phi^{-1}(FAR)]$ where Φ^{-1} represents the inverse of the normal cumulative density function, and HR and FAR represent the hit rate (the proportion of trials with a clockwise average orientation where participants responded clockwise) and false alarm rate (the proportion of trials with a counter-clockwise average orientation where participants responded clockwise), respectively. Note that for our task hits and false alarms are arbitrarily defined. We now include a clarification in the main text (**lines 139-144**) and the equation in the methods section to avoid any confusion about this point (**lines 649-660**).

Fig 2B: Are the fits to the psychometric curve per-subject and averaged, or direct fits of the group average? Can you add "degrees" labels to the horizontal axes?

The fits were carried out on the average (across participants) proportion of clockwise responses but they were performed for illustration purposes only. All key analyses were carried out using our bias index measures on a per-subject (and per-condition) basis. We now clarify this point in the figure legend. We have also added axis labels as requested by the reviewer.

Fig 3B: How can choice accuracy drive confidence? Choice accuracy is not known to the

subject before confidence and thus cannot directly determine confidence. Is it supposed to be a proxy for some unobserved latent variable in the subject's head?

We included choice accuracy to capture trials where subjects “knew” that they made an error (e.g. pressed wrong button), building on the intimate link between confidence and error detection (Yeung & Summerfield 2012 Philosophical Transactions of the Royal Society). We now clarify this point in the main text (**lines 196-197**).

Furthermore, how do prior cues impact confidence? Do they change confidence ratings in the expected way?

The regression analyses presented in Fig. 5D addresses this question. Overall, the effect is in the expected direction, with the prior cue increasing confidence. Interestingly, the effect of the prior cue on confidence decreases with high-contrast (because choices and confidence are driven mainly by the stimulus), but it also decreases with high-variability (again, because people “believe” they are more certain about the stimulus). We now directly make these points (**lines 313-315**).

Fig 5: shows only the model comparison between two models. What happened to the third model that was fitted to the subjects' data?

We have now added the predictions of the variability-mixer model (previously the variability-blind model) to the figure, hoping this aids interpretation of our results.

Fig 6D: why does blocking results in a higher bias index for the low-c stimuli?

This difference could be interpreted as having a “higher bias index when blocking contrast” or as having a “lower bias index when blocking variability”. To understand this result, it is useful to keep in mind the difficulty of the condition that accompanies the low-c condition in the two blocking conditions. When contrast is blocked, the low-c condition is accompanied by the hardest condition (low contrast, high variability), but when variability is blocked, the low-c condition is accompanied by the easiest condition (high contrast, zero variability). We therefore believe that the difference in bias index reflects an overall shift in the bias index across blocks: participants relied more heavily on the cue in the most difficult block. We now include this clarification in the main text (**lines 354-359**).

I331: "and therefore, unlike an optimal model, adopt a common choice threshold across trials" - this model is also optimal, but for different underlying assumptions. In fact, those assumptions are more realistic even for the experiments that are described in the manuscript (see above).

We hope the revisions concerning modelling described addresses this issue.

I345: "Under the subsampling account, ..." - I don't understand this argument. Isn't the variability of individual orientations the same, independent of if there are 4 or 8 on the screen? If so, wouldn't subsampling a single orientation yield the same amount of variability, independent of the number of items on the screen? The same argument would then apply to subsampling up to 4 orientations - the overall variability should be invariant to the number of orientations on the screen.

In our experiments, the distribution of average orientations was constant across all conditions and orthogonal to any other manipulation. In the absence of integration noise, subsampling would not predict the same drop in accuracy irrespective of the number of items on the screen. If, for instance, participants sampled four items, then there would be no mismatch between the average of the sampled items and the true average of the trial (apart from any error due to encoding noise). Instead, for the condition with eight items, there would be a large mismatch between the average of the four sampled items and the true average of the eight items on the screen. By contrast, our integration-noise account predicts that the performance decay is proportional to the variability of the items in the array. Finally, we made sure that the measured

variability across the two set-sizes was the same, by scaling the variance of the distribution from which they are sampled. Without this step, the measured variability of the smaller set-size would be systematically lower and invalidate between-condition comparisons. We have made these points explicit (**lines 854-857**).

Fig S1: "Four pairs of PDFs, one for each of the four conditions in Exp. 3" - weren't there 6 conditions in Exp. 3? 2 contrasts, 3 variability levels?

This was a mistake on our part – thanks for spotting this. We have now made the clarification (**line 882**).

Fig 3C-E: what is the horizontal axis? Is it the average orientation away from zero? If yes, could you provide the degrees?

Thanks for pointing out the lack of clarity in this figure (Fig. S3). The horizontal axis indeed represents the average distance of the average orientation to the category boundary. The space was divided into quantiles (of roughly one degree of width and roughly one degree apart from one another starting at zero degrees). We now include this information in the figure panel (**lines 958-959**).

Reviewer #3 (Remarks to the Author):

The manuscript addresses a central question in the domain of decision-making: the gap between the apparent optimality of perceptual decisions and lack of it in cognitive ones. This is definitely one of the most challenging puzzles facing this field, and any progress towards solving it is important and of high impact for the field. To address the question, the authors have carried out a comprehensive psychophysical and computational study (6 experiments), which manipulate what they refer to as encoding and integration noise, and examined the impact on two dependent variables: i) the reliance on priors (i.e., biasing due to cues), ii) the degree of confidence in the choice. Using an ideal-observer modeling approach, the authors show that while the observers appear to modulate their use of priors and their confidence (as they should) based on encoding-noise, they appear to not do this for the integration-noise. The authors also distinguish between two sub-models that could account for the neglect of the integration-noise and they provide evidence to support the idea that although the observers can discriminate trials based on integration-noise, they still appear to not take this type of noise into account in their decision-process.

I find the study novel and original, very well executed and the paper clearly written. As the question is also of prime importance to the field, I do believe that a revised manuscript, which should clarify few conceptual issues (helping to further strengthen the paper) and a number of more technical ones (allowing the readers to better understand it), would provide an excellent contribution to the journal. Below I list a number of such issues, which the authors should try to address in their revision.

We thank the reviewer for recognizing the value of our work. We are grateful for the suggestions which we have implemented in full.

Conceptual

1. What about-RT?

The data convincingly shows that the amount of biasing and the degree of confidence are modulated by encoding but not by integration-noise (variability of the orientation). However, what about decision-RT? In the Supplement, it is reported that, in fact, the RT is affected by the variability: observers are slower in deciding about stimuli with high variability. The authors use this to argue against a model that predicted faster-RT for high variability trials. This is fine, but the issue is how does the present model account for the slow down. What about the possibility that, in fact, the observers increase the decision-criterion to compensate for the difficulty of the high-variability condition, accounting thus for the slowdown? If that was the case, we would have a situation, in which, while the observers fail to modulate the priors and confidence, they do adjust the decision criterion. This would indicate a more subtle situation than the one (noise-blindness) which the authors support.

Thanks for raising this great point. We would expect an increased decision criterion to result in (1) longer RTs and (2) similar or higher choice accuracy because of more evidence integration. However, in line with previous work (e.g. de Gardelle & Summerfield 2011 PNAS; Li et al 2017 PLoS Computational Biology), we found that, while high orientation variability caused longer RTs, it also caused a decrease in accuracy, in contrast to the prediction of the decision-criterion account. These results are consistent with increased difficulty for combining variable information and were best explained by a decrease in drift rate as shown in the Supplementary Information (**lines 922-961**).

We have modelled these effects in previous work (De Gardelle & Summerfield, PNAS, 2011) where we assume that the feature values are transduced via a compressive nonlinearity – this nonlinearity penalises outliers and in turn leads to a disproportionately lower input signal to the accumulation process. This model successfully captured our previous data, and we expect that it would do so here as well, but we feel that

such modelling (involving a new model with a different functional form) is beyond the scope of the current manuscript.

Finally, one may also argue, that the observers are somewhat rational in not down-modulating their confidence with variability, if they already tried to compensate for this by taking more samples to improve their choice-performance. The latter would be also consistent with the fact that confidence is still modulated, by RT (RT could be an estimate of the criterion), and the question that may be important to address is whether the benefit from the change of criterion is not efficient enough and why. It is possible that there are other answers to these issues or that I missed something important here. However, I believe that clarifying these issues would make the paper much stronger.

Thank you for this suggestion. The reviewer's comment can be separated into two points.

The first is whether it may be rational for participants not to down-modulate their confidence if they have already tried compensating for the increased difficulty of the trial by taking more time to decide. The proper-scoring rule implemented in Experiment 2 (about which participants received detailed instructions) means that the strategy proposed by the reviewer would result in a lower monetary return and is therefore not optimal in the context of our task.

The second point regards the actual consequence for performance that results from failure to down-modulate confidence (e.g. perhaps such failure had not real impact on performance). To address this issue, we computed the gain in accuracy (accuracy gain) achieved by participants when moving from neutral trials (where the prior cue does not favour any category) to biased trials (where the cue provides the opportunity to compensate for performance) for our critical conditions. Overall, participants achieved positive gains: they used the prior cue to compensate for their errors in all conditions. However, if the intuition of the reviewer is right about participants compensating for poorer performance not by lowering their confidence but by other means, then we would also predict equivalent levels of accuracy gain in the low-c and high-v conditions. Instead, the analysis also shows that only participants and our noise-blind model show a smaller accuracy gain when comparing the condition with low contrast against the condition with high variability. These results indicate that participants' lack of adjustment of confidence resulted both in lower performance in the task as well as lower monetary reward. We include the accuracy-gain analysis in the revised manuscript (**lines 1061-1077**).

2. Types of noise.

Across the manuscript, there is some lack of clarity about the nature of the two types of noise. In the introduction (lines 55-64), encoding noise is associated with low visibility stimuli, while integration-noise is associated "with multiple, some-times discordant pieces of information". So what about a moving dot stimuli array, in which the coherence of the motion varies in time? (should the noise here be mapped to encoding or to integration?). The authors mention in the same sentence (line 63) the possible limitations due to a capacity limited system (this also comes in the Discussion, lines 381-382). Is this the important distinction? But this would seem to go against the result that the results of Exp-6, which indicate that the capacity limitations are not playing much role in the task (?).

In the revised manuscript we have tried to clarify the definitions of encoding and integration noise and how they may be identified in other tasks. We also argue that these two sources of noise may in fact be confounded in several tasks. For instance, we believe that both encoding noise and integration noise are affected by motion coherence in a random dot-motion task (see below). These points are made in the revised Discussion (**lines 424-433** and **437-440**).

A typical random dot-motion (RDM) task includes two sources of noise that are often confounded. First, there is external noise (which reduces the informativeness of the encoded stimulus) in the sense that, even if a single sample of momentary evidence were processed perfectly, accuracy would not be at ceiling because there is a mismatch between the instantaneous evidence that is provided to participants and the

true motion direction. This type of noise would map onto our definition of encoding noise, in the sense that the encoded stimulus already bears uncertainty about the correct category of the trial. Second, as noted by the reviewer, the evidence received by the participant has to be integrated across time. While averaging more samples reduces the first type of uncertainty (encoding noise), it also opens the door for inaccuracies of integration (integration noise). The two sources of noise are confounded because reducing the coherence simultaneously lowers i) the correspondence between instantaneous evidence and the true direction (i.e. there is higher encoding noise), and ii) increases the variability of evidence across time (i.e. there is higher integration noise).

We indeed believe that a reason why more variable items may be harder to integrate is because of a system's capacity limit. However, we do not believe that this necessarily contradicts the results we report in Experiment 6, as the capacity limit need not be one based on the number of items that need to be integrated but rather on how variable they are. For instance, other experiments in our laboratory and elsewhere (e.g. Drugowitsch and colleagues) show increased levels of difficulty for more variable stimuli that occur in temporal succession.

In general, I feel that the paper would benefit from a more clear discussion of the nature of types of noise that the observers have access to and use, and the ones they do not. While this is clear enough for the present study, it is not so for how we should think of generalizing the results to other situations.

We think that any situation that requires people to actively combine information from distinct sources would be susceptible to integration noise, and that people (by virtue of their blindness to this noise) would tend to show departures from what would intuitively be expected under optimal inference. We now include these general points in our discussion (**lines 431-433**), in addition to the revisions detailed above.

A related issue that may also be important to clarify, in this context: I assume that the average orientation in the high-v condition is constrained to be at a fixed value (same as for low-v trials) and this is why the authors refer to this integration-noise as internal (lines 66-67). Perhaps this detail appears in the Suppl, but it is an important one, which should be mentioned in the main text.

Yes, when creating our stimuli, we made sure that the average orientation of a trial was orthogonal to any other manipulation. We now explicitly make this point when describing our task in the Results section (**lines 91-92**).

Minor & clarifications (in order of appearance in the manuscript)

Fig. 1E. Last panel on the right: this appears the same as the previous one (to its left); should this not refer to VARIABILITY instead of contrast?

We only show an example of a contrast trial in Fig. 1E because of space limitations. To avoid confusion, we have updated the legend to read: "*In Experiment 4, after having made a choice, participants were required to categorise (low versus high) either the contrast or the variability of the stimulus array. We show an example 'contrast' trial*" (**lines 132-134**).

Line-139: could be helpful to clarify that the "bias-index" corresponds to CCW-CCCW (it took some time to guess this).

We now clarify this point in the main text (**lines 139-144**) and include a reference to the methods section where additional details are provided (**lines 649-660**).

Lines 178-181: Is the confidence affected by contrast, once RT is factored out?

Yes, in Fig. 3B, we show the regression coefficients for all predictor variables, with contrast (c) having a positive effect on confidence (Fig. 3B second bar), after controlling for RT. We now mention this result (**lines 186-190**).

Line 205-6: “probability of opting in did not vary with variability”. Fig 3D seems to show a variability effect.

Yes, you are right. There is a modest decrease in the probability of opting-in when analysing the conditions directly. However, our ANOVA of the full dataset did not identify a significant effect of variability on opt-in behaviour, neither did our logistic regression analysis of trial-to-trial opt-in responses. We therefore think that the modest decrease may be secondary to other factors. To clarify this point, we have adjusted this sentence to read “the probability of opting in varied with contrast [...] but not with variability [...], after controlling for other task-relevant factors (e.g., average orientation and RTs)” (**lines 214-217**).

Fig. 4F. The light-gray curve is hardly visible. From what I see, however, this curve is the steepest (thus has the best discriminability; why should the noise-blind model have higher discriminability than the optimal one?). Also the white dots in Fig. 4C-E are quite hard to see.

We apologise for the lack of clarity. These curves (Fig. 4F) represent “belief functions” for different models (with similar levels of noise but different beliefs as detailed in the text). The steeper curve for the noise-blind model therefore does not indicate higher accuracy but instead a stronger (even if less accurate) belief about the category membership of a stimulus. We have revised the legend for this figure and included figure legends to avoid any confusion (**lines 265-267**).

Lines 262-279. It could help to clarify the differences in the predictions between the two non-optimal models (if possible, illustrate them in a plot). Furthermore, it is not clear enough, how any of the models are used to make predictions about confidence. In particular, it will be important to explain how the models are used to create the Figures in Fig. 5A. What determines the variability we see (in the models)? Perhaps it could also be more efficient to start this paragraph with the Bias-index, as this was discussed more earlier on, before shifting to the confidence, and explaining this in more detail.

We have added panels to the figure, including the predictions for the variability-mixer (formerly variability-blind) model (see new Fig. 5, Fig. S2 and Fig. S6). Model confidence is computed directly from a model’s posterior beliefs about the stimulus categories given sensory evidence and prior cue (see equation 4, **line 577**). In other words, confidence is the ratio between the posterior belief about the chosen category, and the sum of the posterior beliefs about the chosen and the unchosen categories. We note that within-model variability is due to participant variability (i.e. the amount of encoding and integration noise differs between participants). We now make these points in the main text (**lines 294-295**).

Line-323: “but” missing (I think) before “only...”

Thank you for spotting this mistake.

Lines 337-338. “We have proposed that orientation variability impairs performance because of additional noise inherent to cognitive integration of variable or discordant pieces of information. However, another possible explanation of this relationship is that participants based their choices on a subset of gratings rather than the full array”. It is not clear, in fact, why this is ANOTHER explanation (?). If the integration-noise is due to the capacity limitations of WM or of the attentional system, than the second sentence is quite congruent with the first one.

Thank you for pointing this out. You are right that subsampling could in principle be the cause of integration noise. However, we believe that this is not the case. We acknowledge that our arguments against subsampling, and the distinction between subsampling and integration noise, had been unclear and

scattered. We have therefore included a new section in the Supplementary Information which summarises our arguments against subsampling and clarifies the specific predictions that a subsampling account makes about response patterns.

Reviewers' Comments:

Reviewer #1:

Remarks to the Author:

The authors have addressed satisfactorily most of my comments, except for one.

About my question whether the task really distinguishes sensory from cognitive noise, I do not fully agree with the answer by the authors. In principle, I agree about the advantage of a single task in which sensory and cognitive variability could be manipulated separately. Yes, it is reasonable to assume that manipulating contrast vs orientation variability will affect different computations and the related neural circuits, such as "encoding" vs "integration" to use the authors terminology. But I still cannot understand how the proposed experiments rule out that the "integration" noise effects are not purely in the sensory domain, e.g. originating due to circuit interactions within the early visual cortex, where spatial (and temporal) interactions are abundant and well documented. The effects observed in the "integration" case are qualitatively similar to those observed in purely cognitive tasks, but perhaps one could consider the alternative explanation that sensory processing is near-optimal only for some stimulus dimensions (contrast) and not others (spatio-temporal integration), and that in the latter case it fails in ways similar to those observed in purely cognitive tasks; can this be ruled out in the experiments reported here? For instance, I may have missed an indication of the spatial size and location of the grating patches in Methods, but if they span both hemifields as Fig. 1 seems to suggest, the sensory integration sub-optimality may reflect sensory optimization to natural scene statistics: in the natural environment, visual inputs are rarely homogeneous across such a large portion of the visual field. Would the experimental results change, and how, if the gratings array were presented within a single hemifield and altogether covered roughly the average V2 or V4 receptive field size? On a related note, the reference to Michael et al 2015, Cerebral Cortex, and related text in the rebuttal, may be helpful in the manuscript.

So, again, the results are sound as far as I understand, but I am not sure they address sensory vs cognitive inferences, as opposed to two separate sensory computations. Given the centrality of "cognitive inference" to this manuscript, I think this is not simply a terminology issue, but more of a scope issue.

Reviewer #2:

Remarks to the Author:

The revised manuscript is significantly improved in clarity, and now makes a much stronger case that it is indeed noise blindness that drive suboptimal choices. In particular, the authors provided convincing arguments that subsampling is not a good alternative hypotheses to explaining the observed choice suboptimality. What I previously didn't appreciate is that the mean orientation was held constant while reducing the number of orientations, which would lead to a larger choice variability with subsampling 8 orientations. This, and other points, is now clarified in the manuscript. Overall, I have one single remaining concern about the manuscript that involves the different models and how they are fit to the data.

Specifically, the authors consider their omniscient model optimal (as they call the two other models "suboptimal", see l234). This model assumes that the participants know the contrast and variability in each trial, and use this knowledge to make choices. However, as trials were interleaved, they could not know this information before onset of the stimulus. They could infer these quantities from the stimulus itself, but they would need to do so at the same time as identifying the stimulus category (CW or CCW). In particular variability would use the same information as the category judgment,

namely the perceived orientation of the individual oriented pattern. Therefore, it is odd to assume perfect condition inference, while assuming imperfect category inference. Instead, a realistic (i.e., non-omniscient) optimal model would explicitly infer both at the same time. This is different from the noise-blind or variability-mixer models, which are indeed suboptimal. Nonetheless, it might at least qualitatively replicate some (but probably not all) of the effects that the authors attribute to noise blindness. This definitely needs to be clarified before the work can be published.

To avoid any further misunderstandings, let me spell out explicitly what I consider to be a realistic optimal model. When doing so, I am ignoring bias cues, but they can be easily integrated into the model. Let x be the average orientation, drawn according to category cat in $\{\text{CW}, \text{CCW}\}$ according to $p(x | \text{cat})$ (as specified in the manuscript). Furthermore, let z_1, \dots, z_K be shown the orientation of the stimulus ($K=8$ for 8 orientations). We assume

$$(R1) z_k | x \sim N(x, \text{sig2var}),$$

where sig2var is the variance that determines the variability (e.g., $\text{sig2var} = 0$ for no variability). The participants observe y_1, \dots, y_K , which are noisy instantiations of z_1, \dots, z_K , with

$$(R2) y_k | z_k \sim N(z_k, \text{sig2cont}),$$

where the variance sig2cont is determined by the contrast (e.g., low contrast yields high sig2cont). The condition cond determines sig2var and sig2cont . Overall, if we marginalize over the z_k 's, we get

$$(R3) y_k | x \sim N(x, \text{sig2var} + \text{sig2cont}) \sim p(x, \text{cond}).$$

The participants observe y_1, \dots, y_K and want to identify the category cat . That is, they want to find $p(\text{cat} | y_1, \dots, y_K)$. To do so, we first find the joint over x , cond , and cat , given by

$$(R4) p(\text{cat}, \text{cond}, x | y_1, \dots, y_K) = p(y_1, \dots, y_K | x, \text{cond}) p(x | \text{cat}) p(\text{cond}) p(\text{cat}) / Z,$$

where Z is the adequate normalization constant. As neither x nor cond are known to the participants (unlike in the manuscript's omniscient model), we need to marginalize over them, resulting in

$$(R5) p(\text{cat} | y_1, \dots, y_K) = \sum_{\text{cond}} \int_x dx p(y_1, \dots, y_K | x, \text{cond}) p(x | \text{cat}) p(\text{cond}) p(\text{cat}) / Z,$$

which is the optimal posterior from the participant's perspective. In particular, CW will be chosen if $p(\text{cat} = \text{CW} | y_1, \dots, y_K) > 0.5$. This is a different expression from the manuscript's variability-blind model, as it doesn't assume that participants can't infer the variability. Rather, it assumes that the variability is unknown (but might differ across trials - hence a prior $p(\text{cond})$), and needs to be marginalized out (as done in Deneve (2012); Drugowitsch et al. (2012); van der Berg & Ma (2012) <- the last one is certainly known to the Summerfield lab). Nonetheless, subjects could infer this variability by computing, for example, $p(\text{cond} | y_1, \dots, y_K)$.

The experimenter doesn't directly observe the y_1, \dots, y_K 's but instead only the z_1, \dots, z_K . Then, choice probabilities can be identified as follows. Let $f(y_1, \dots, y_K) = 1$ if $p(\text{cat} = \text{CW} | y_1, \dots, y_K) > 0.5$, and $f(y_1, \dots, y_K) = 0$ otherwise. The predicted probability of choosing CW is then given by

$$(R6) p(\text{choose CW} | z_1, \dots, z_K, \text{cond}) = \int f(y_1, \dots, y_K) p(y_1, \dots, y_K | z_1, \dots, z_K, \text{cond}) dy_1, \dots, y_K .$$

The confidence would differ from this probability, and is instead given by $p(\text{cat} = \text{choice} | y_1, \dots, y_K)$. In particular, due to marginalizing over cond , participants would be seemingly underconfident in easy

conditions and overconfidence in hard conditions. This is not a suboptimality, but arises because Eq. (R6) conditions on cond, which is unknown to the participant and thus leads to this deviation (this point is elaborated in a convoluted way in Drugowitsch, Moreno-Bote & Pouget (2014)).

Overall, this would constitute the optimal model without assuming omniscience. A variant thereof could assume knowing the contrast (as this can't be inferred from y_1, \dots, y_K , but requires other stimulus features) while still marginalizing over different variability levels. With such marginalization, the statement "the model makes the same predictions about choice on neutral trials" (I602) would likely not be true anymore.

Another concern about the model fits is how choice probability predictions are computed. The manuscript only states that "Our modeling approach allowed us to calculate choice probability [...]" (I604), but does not elaborate on how this is performed. I have frequently observed in published work that the posterior, in this case Eq. (3) in the manuscript, is mistaken for the choice probability. As the authors correctly point out, this posterior instead triggers choices deterministically (I575/576). Stochastic choices then emerge by marginalizing over the various noisy variables that the posterior depends on, as I have done above in Eq. (R6). Is this also what the authors are doing to fit their models to behavior? This needs to be clarified.

Minor comments:

I157: "Taken together, these results show that participants did not adapt to the additional noise arising during integration of discordant pieces of information". Would the interaction between cue and variability in Fig. 2E suggest that such adaptation seems to occur? Furthermore, shouldn't the bias index in Fig. 2D be independent of v for true variability blindness?

I181: "it did not vary with variability" - analysis is based on means. Other moments might vary with variability.

Fig 3B: why were only bias trials used for this analysis?

I204: "more direct measure, of" -> "more direct, measure of"

I329: "identifying the variability condition but otherwise" - "were" missing

Fig S2, caption: "men" -> "mean"

Reviewer #3:

Remarks to the Author:

I find the manuscript significantly improved, with regards to additional helpful analyses (for example the sub-sampling and the sequential-sampling models) and a clearer presentation and discussion. The authors also provided reasonable answers to most of my queries and I believe that they touch on an important distinction (encoding vs integration noise) for the decision-optimality debate. There are still a few issues, which I believe that could be further clarified and discussed, in order to obtain an effective contribution for this journal. I will elaborate on these issues below, starting from the more conceptual to the technical ones.

1. How does increased-variability affect RT?

The authors have responded to my query of whether participants act upon increased variability in a different way from reduced contrast, by increasing the response-criterion, so as to require more evidence samples compensating for difficulty, by stating that while such a variation would explain the increase in RT it should keep accuracy at a fixed value. This would be indeed the case if the change in criterion is perfectly calibrated to compensate for the increase in difficulty. However, it is possible that the change of criterion is partial, resulting in both an increase in RT and a reduction in accuracy. I am not saying that this is the case, and I also accept that the authors' proposal that increased variability leads to a reduced drift (due to evidence-compression caused by outlier-neglect) is a reasonable one. It would be good, however, to see this shown in the sequential-sampling simulation. As presented now, this simulation only disproves the fixed-threshold, equal-drift/increased-drift-variability model. It should be possible, however, to extend this model analysis to also contrast between the increased-criterion hypothesis and the reduced-drift one. To do this all one has to do is to include these two models (which make different assumptions on how increased variability) affects the diffusion parameters, and see via BIC-measures, which provides a better account to the data. One technical point here is the fit measure that is being used in model fitting and comparison. I believe that this cannot be based on accuracy scores only, because doing so gives a high weight to accuracy and neglects RT (and then it is not very surprising if the RT fit is less good); also a difference in accuracy between .95 and 1 is not the same as one between .75 and .8, due to ceiling compression effects. Rather this should be based on Log-Likelihood for a response to be correct/incorrect and within a certain time bin (this can be also done using quantile-RTs; Ratliff & McKoon, 2008). I believe that such an exercise would significantly strengthen the conclusions (whatever they are) without requiring the more complicated model, based on outlier-neglect. I agree that the latter does not need to be simulated in this paper, but I think that it could be helpful to mention it in the Discussion as a reason for the drift reduction (if this is what the model comparison shows; in case there is a tie between drift-reduction and the criterion-increase, this can also be discussed). I agree that this is secondary to the issue of why variability has a different effect on confidence and on weights given to priors, however, RT is an important aspect of the decision-process, having implications for the issue of whether participants totally discard integration noise, or only partially do so).

2. Encoding- vs. integration-noise and limited capacity.

I am still confused about how we should understand the integration noise and its relation to capacity limitations. The authors stated in response to my query that the capacity limitation does not involve a fixed number of items, but rather the degree of variability. But this is exactly what we need to explain. It can help to make this more concrete by considering a mechanism-integration example. Consider, the LIP-accumulators, which were proposed to carry out evidence-integration in decision tasks (Gold, Shadlen, et al). The way we typically understand such a mechanism is that its advantage is exactly to do away with the capacity limitation of the WM-system. This is because one only keeps a single (or two) integrated tally (ies) of the evidence, rather than all the samples which would exceed the capacity limitation of the WM system. So, the question that would be helpful to discuss is, what do these results imply about such an integration mechanism? Few options one may consider is that the mechanism is subject to some deficiency (blinks, or recency/primacy of temporal weighting) that give rise to what is here called "integration-noise" (or perhaps that such a mechanism is not really available at all). In addition, it may be helpful to discuss whether such integration-deficiency (or noise) apply equally to temporal integration (as in the regular moving dot paradigm) or is particular to integration across space (as in this paper).

Technical points

p. 3 (1st paragraph). What about optimality in the SPRT sense (Gold & Shadlen, 2001; Bogacz et al., 2006), which seems to involve temporal-integration of evidence. Should this not be included in the optimal part of the decision mechanism?

- p. 3. Last paragraph. Why is the integration of multiple pieces of evidence difficult? (from the perspective of a LIP-integrator, it should not be?)
- p. 6. The bias-index shows a positive main effect of contrast (both text and D-caption). Should this not be "negative" main effect: less bias for high contrast?
- p. 9. Fig. 4. It may be helpful to explain why the noise-blind functions appear more skewed (I assume this is due to the fact that the noise is convolved with the red-blue truncated distributions in panel-A).
- p. 10, I find the correlation-data in Fig-4 A, less significant (and more difficult to follow) for the argument than the results presented in the Supplement (S2) about the over-confidence. So, you may like to consider if it is not better to swap those.
- p. 11. Line 306. "Models differ". (which models?)
- p. 11 Line 329. "but otherwise aware". (are aware?)
- p. 29. Sequential-Sampling Suppl. What type of drift variability is used. Is this variability between trials (or within a trial)?

Reviewer #1 (Remarks to the Author):

The authors have addressed satisfactorily most of my comments, except for one.

We thank the Reviewer for the positive assessment of our work. We have addressed their remaining concern below – which we hope the Reviewer will find satisfactory.

About my question whether the task really distinguishes sensory from cognitive noise, I do not fully agree with the answer by the authors. In principle, I agree about the advantage of a single task in which sensory and cognitive variability could be manipulated separately. Yes, it is reasonable to assume that manipulating contrast vs orientation variability will affect different computations and the related neural circuits, such as “encoding” vs “integration” to use the authors terminology. But I still cannot understand how the proposed experiments rule out that the “integration” noise effects are not purely in the sensory domain, e.g. originating due to circuit interactions within the early visual cortex, where spatial (and temporal) interactions are abundant and well documented. The effects observed in the ‘integration’ case are qualitatively similar to those observed in purely cognitive tasks, but perhaps one could consider the alternative explanation that sensory processing is near-optimal only for some stimulus dimensions (contrast) and not others (spatio-temporal integration), and that in the latter case it fails in ways similar to those observed in purely cognitive tasks; can this be ruled out in the experiments reported here? For instance, I may have missed an indication of the spatial size and location of the grating patches in Methods, but if they span both hemifields as Fig. 1 seems to suggest, the sensory integration sub-optimality may reflect sensory optimization to natural scene statistics: in the natural environment, visual inputs are rarely homogeneous across such a large portion of the visual field. Would the experimental results change, and how, if the gratings array were presented within a single hemifield and altogether covered roughly the average V2 or V4 receptive field size? On a related note, the reference to Michael et al 2015, Cerebral Cortex, and related text in the rebuttal, may be helpful in the manuscript.

So, again, the results are sound as far as I understand, but I am not sure they address sensory vs cognitive inferences, as opposed to two separate sensory computations. Given the centrality of ‘cognitive inference’ to this manuscript, I think this is not simply a terminology issue, but more of a scope issue.

We appreciate this concern, and the reviewer is right to raise it. In the revised submission, we now include an extended discussion of this issue. We accept that using our behavioural paradigm alone it is hard to categorically tell whether the locus of “integration noise” is early (e.g., in early sensory cortex) or late (e.g., at a more “cognitive” level, such

as in higher association cortex where information is combined with broader windows in space and time). Below, we outline our argument for why the locus of integration is more likely to be late than early, whilst accepting that further research is needed to test this hypothesis. Our argument rests on 4 points:

- (1) Participants can detect stimulus variability (Experiment 4), which we would not expect if the sensory representations themselves were distorted (e.g., due to circuit interactions among orientation columns in primary visual cortex), and in support of a higher-order account, the observed noise blindness does not depend on whether stimulus variability was accurately discriminated (Fig. 6A-B)
- (2) The effect of stimulus variability on subject's choice accuracy does not depend on set size (4 versus 8 items in Experiment 6), which we would not expect if 'integration' noise was entirely by constraints on early sensory processing, and again in support of a higher-order account, the observed noise blindness did not depend on set size.
- (3) Participants' response times and drift-diffusion modelling indicate that information processing continues after stimulus offset – processing which is unlikely to happen in early sensory areas.
- (4) As the reviewer notes, fMRI evidence from our own lab (Michael et al., 2013, *Cerebral Cortex*) shows that stimulus variability is reflected in activity in 'association' areas (parietal cortex) and 'control' areas (anterior insula and dorsomedial prefrontal cortex), which is consistent with 'integration' noise arising after sensory encoding.

In summary, while we cannot rule out that 'integration' noise is partly of a sensory origin, we believe that the evidence overall indicates that the source of 'integration' noise is mainly in the cognitive domain.

Here is the relevant paragraph from the main text:

"We recognize that using our task alone it is hard to categorically say whether the locus of integration noise is early (e.g., in early sensory cortex) or late (e.g., higher association cortex where information is combined with broader windows in space and time). However, several lines of evidence indicate a late locus. First, participants can detect stimulus variability (Exp4), which we would not expect if the sensory representations themselves were distorted during early processing stages, and in support of a higher-order account, noise blindness does not depend on whether variability was accurately discriminated (**Fig. 6A-B**). Second, the effect of stimulus

variability on subject's choice accuracy does not depend on set size (four vs eight items in Exp6), and again in support of a higher-order account, noise blindness did not depend on set-size. Third, participants' response times and drift diffusion modelling suggest that information processing continues after stimulus offset and is thus unlikely to occur in early sensory areas. Fourth, stimulus variability is reflected in brain activity in higher association and control areas (e.g., parietal cortex, anterior insula and dorsomedial prefrontal cortex), which is consistent with integration noise arising after sensory encoding (Michael, Gardelle, Nevado-Holgado, & Summerfield, 2013). More broadly, our study is an example of the now conventional approach of using perceptual paradigms as a window onto general principles of cognition and decision making (Shadlen & Kiani, 2013)." (Discussion, **lines 461-476**)

Reviewer #2 (Remarks to the Author):

The revised manuscript is significantly improved in clarity, and now makes a much stronger case that it is indeed noise blindness that drive suboptimal choices. In particular, the authors provided convincing arguments that subsampling is not a good alternative hypothesis to explaining the observed choice suboptimality. What I previously didn't appreciate is that the mean orientation was held constant while reducing the number of orientations, which would lead to a larger choice variability with subsampling 8 orientations. This, and other points, is now clarified in the manuscript.

We thank the Reviewer for the positive assessment of our work.

Overall, I have one single remaining concern about the manuscript that involves the different models and how they are fit to the data. Specifically, the authors consider their omniscient model optimal (as they call the two other models "suboptimal", see l234). This model assumes that the participants know the contrast and variability in each trial, and use this knowledge to make choices. However, as trials were interleaved, they could not know this information before onset of the stimulus. They could infer these quantities from the stimulus itself, but they would need to do so at the same time as identifying the stimulus category (CW or CCW). In particular variability would use the same information as the category judgment, namely the perceived orientation of the individual oriented pattern. Therefore, it is odd to assume perfect condition inference, while assuming imperfect category inference. Instead, a realistic (i.e., non-omniscient) optimal model would explicitly infer both at the same time. This is different from the noise-blind or variability-mixer models, which are indeed suboptimal. Nonetheless, it might at least qualitatively replicate some (but probably not all) of the effects that the authors attribute to noise blindness. This definitely needs to be clarified before the work can be published.

We thank the Reviewer for taking time to spell out so carefully the proposed 'realistic' optimal model. We agree that it is important to test behaviour under this model and have done so in simulation, with minor modifications required by our task design, as detailed below. In summary, (a) the model does not predict a performance cost for high-variability stimuli, which further supports that 'integration' noise does not arise during early sensory processing and warrants our inclusion of a second noise term which scales with stimulus variability, and (b) the model predicates a pattern of confidence opposite to that shown by participants (and our noise blind model). Specifically, while accuracy is independent of stimulus variability, confidence drops for high-contrast, high-variability stimuli, because the condition is "confused" with the conditions of lower contrast. These results further support our argument that noise blindness is due to a lack of awareness of the additional noise introduced by stimulus variability. We introduce this model in the main text – where

we motivate the model as a control for the ‘simplifying’ assumptions of our other models – and describe the model in full (**lines 943-1046**) and report the simulation results (Fig. S2) in the Supplementary Information.

Here are the relevant paragraphs from the main text:

“In our task the distribution of average orientations was common across experimental conditions and consequently independent of contrast and variability (**Fig. 4A**). We therefore modelled an agent’s sensory evidence as a random (noisy) sample from a Gaussian distribution centred on the average orientation of the stimulus array (**Fig. 4B**), with the variance of this distribution determined by both encoding noise and integration noise. In the Supplementary Information, we show that the results reported below are not due to this simplifying assumption (see **Fig. S2** where we simulate data under a Bayesian model which operates with eight noisy samples, one for the orientation of each grating, rather than one noisy sample).” (Results, **lines 237-244**)

“In the aforementioned models, we assumed that an observer’s inferences are conditional on the average orientation of a stimulus array. We made this simplifying assumption because, by design, the average orientation is independent of variability in the orientation of individual items across experimental conditions. However, it is possible that, if we modelled an observer’s inferences as conditional on the orientation of individual items, then integration noise may not be needed to account for the performance cost associated with high-variability stimuli. In the Supplementary Information, we simulate performance under this ensemble model, and show that it cannot predict the performance cost associated with high-variability stimuli (**Fig. S2**)” (Methods, **lines 632-639**):

To avoid any further misunderstandings, let me spell out explicitly what I consider to be a realistic optimal model. When doing so, I am ignoring bias cues, but they can be easily integrated into the model. Let x be the average orientation, drawn according to category cat in $\{CW, CCW\}$ according to $p(x | cat)$ (as specified in the manuscript). Furthermore, let z_1, \dots, z_K be shown the orientation of the stimulus ($K=8$ for 8 orientations).

Modification 1: We did not simply sample the individual items from a distribution with mean x and variance $sig2var$; sampling the items in this way would have introduced a confound between $sig2var$ and “boundary difficulty” (i.e. the average unsigned difference between the category boundary and the actual mean of the sampled orientations, $z_1 \dots z_K$). We therefore included a ‘correction’ under which the sampled z ’s first had their mean removed and then had the pre-specified x added to the mean-subtracted value. This correction ensures that the pre-specified x (sampled from a common distribution for all conditions) would not be affected by stimulus variability and thereby removes the potential confound between $sig2var$ and boundary difficulty.

We assume

$$(R1) z_k | x \sim N(x, \text{sig2var}),$$

where sig2var is the variance that determines the variability (e.g., $\text{sig2var} = 0$ for no variability). The participants observe y_1, \dots, y_K , which are noisy instantiations of z_1, \dots, z_K , with

$$(R2) y_k | z_k \sim N(z_k, \text{sig2cont}),$$

where the variance sig2cont is determined by the contrast (e.g., low contrast yields high sig2cont). The condition cond determines sig2var and sig2cont . Overall, if we marginalize over the z_k 's, we get

$$(R3) y_k | x \sim N(x, \text{sig2var} + \text{sig2cont}) \sim p(x, \text{cond}).$$

The participants observe y_1, \dots, y_K and want to identify the category cat . That is, they want to find $p(\text{cat} | y_1, \dots, y_K)$. To do so, we first find the joint over x , cond , and cat , given by

$$(R4) p(\text{cat}, \text{cond}, x | y_1, \dots, y_K) = p(y_1, \dots, y_K | x, \text{cond}) p(x | \text{cat}) p(\text{cond}) p(\text{cat}) / Z,$$

where Z is the adequate normalization constant. As neither x nor cond are known to the participants (unlike in the manuscript's omniscient model), we need to marginalize over them, resulting in

$$(R5) p(\text{cat} | y_1, \dots, y_K) = \sum_{\text{cond}} \int_x dx p(y_1, \dots, y_K | x, \text{cond}) p(x | \text{cat}) p(\text{cond}) p(\text{cat}) / Z,$$

which is the optimal posterior from the participant's perspective. In particular, CW will be chosen if $p(\text{cat} = \text{CW} | y_1, \dots, y_K) > 0.5$. This is a different expression from the manuscript's variability-blind model, as it doesn't assume that participants can't infer the variability. Rather, it assumes that the variability is unknown (but might differ across trials - hence a prior $p(\text{cond})$), and needs to be marginalized out (as done in Deneve (2012); Drugowitsch et al. (2012); van der Berg & Ma (2012) <- the last one is certainly known to the Summerfield lab). Nonetheless, subjects could infer this variability by computing, for example, $p(\text{cond} | y_1, \dots, y_K)$.

The experimenter doesn't directly observe the y_1, \dots, y_K 's but instead only the z_1, \dots, z_K . Then, choice probabilities can be identified as follows. Let $f(y_1, \dots, y_K) = 1$ if $p(\text{cat} = \text{CW} |$

$y_1, \dots, y_K) > 0.5$, and $f(y_1, \dots, y_K) = 0$ otherwise. The predicted probability of choosing CW is then given by

$$(R6) \ p(\text{choose CW} \mid z_1, \dots, z_K, \text{cond}) = \int f(y_1, \dots, y_K) \ p(y_1, \dots, y_K \mid z_1, \dots, z_K, \text{cond}) \ dy_1, \dots, y_K.$$

The confidence would differ from this probability, and is instead given by $p(\text{cat} = \text{choice} \mid y_1, \dots, y_K)$. In particular, due to marginalizing over *cond*, participants would be seemingly underconfident in easy conditions and overconfidence in hard conditions. This is not a suboptimality, but arises because Eq. (R6) conditions on *cond*, which is unknown to the participant and thus leads to this deviation (this point is elaborated in a convoluted way in Drugowitsch, Moreno-Bote & Pouget (2014)).

Modification 2: Because the aforementioned ‘correction’ makes it difficult to derive model predictions analytically, we quantified model behaviour by averaging performance over several noisy instantiations (samples) of $y_1 \dots y_K$.

Overall, this would constitute the optimal model without assuming omniscience. A variant thereof could assume knowing the contrast (as this can't be inferred from y_1, \dots, y_K , but requires other stimulus features) while still marginalizing over different variability levels. With such marginalization, the statement "the model makes the same predictions about choice on neutral trials" (I602) would likely not be true anymore.

Modification 3: We limited our model simulations to neutral (unbiased) trials. These trials already allowed us to conclude that the model (which we call the *ensemble* model) cannot account for the performance cost associated with stimulus variability and does not predict overconfidence for high-variability stimuli. We also did not extend the model to mimic our noise-blind and variability-mixer models. While we agree that it is more ‘realistic’ for a model to operate with the orientations of individual items in a stimulus array, we believe that our current modelling approach, despite the ‘simplifying’ assumption that a model operates with the mean orientation of a stimulus array, still conveys the key message that (a) ‘integration’ noise is an additional source of noise and (b) that blindness to this noise, and not simply being unable to attribute the noise to an appropriate cause, is what drives sub-optimality in our data. However, as stated above, we acknowledge the ‘simplifying’ assumption of our models in the main text and refer the reader to the Supplementary Information for a more realistic model.

Another concern about the model fits is how choice probability predictions are computed. The manuscript only states that "Our modeling approach allowed us to calculate choice probability [...]" (I604), but does not elaborate on how this is performed. I have frequently

observed in published work that the posterior, in this case Eq. (3) in the manuscript, is mistaken for the choice probability. As the authors correctly point out, this posterior instead triggers choices deterministically (I575/576). Stochastic choices then emerge by marginalizing over the various noisy variables that the posterior depends on, as I have done above in Eq. (R6). Is this also what the authors are doing to fit their models to behavior? This needs to be clarified.

We confirm that choices were a deterministic function of the posterior estimates (i.e., the category with the highest posterior was always chosen), and that choice probabilities were obtained by marginalizing over different instantiations of sensory evidence. For the ‘original’ models, where inferences are conditional on the average orientation of a stimulus array, choice probabilities are easily computed. For the ensemble model, where inferences are conditional on the individual items of a stimulus array, choice probabilities were estimated by averaging over different instantiations of sensory evidence. We now clarify how choices were computed in the Methods (**lines 628-631**).

Minor comments:

I157: "Taken together, these results show that participants did not adapt to the additional noise arising during integration of discordant pieces of information". Would the interaction between cue and variability in Fig. 2E suggest that such adaptation seems to occur? Furthermore, shouldn't the bias index in Fig. 2D be independent of v for true variability blindness?

The ‘adaptation’ identified by the reviewer (Fig. 2E) goes in the opposite direction of what one would intuitively expect: the negative interaction effect between cue and variability suggests that people rely less on the cue in high-variability trials. With respect to the bias-index decreasing with stimulus variability (Fig. 2D), it is important to note that we obtained the same qualitative pattern under the noise blind model (Fig 5C).

I181: "it did not vary with variability" - analysis is based on means. Other moments might vary with variability.

We now acknowledge this point in the main text:

“In support of our hypothesis, analysis of the full factorial design showed that, while mean confidence (our analyses pertain to the mean and not other moments) varied with contrast...” (Results, **lines 180-182**).

Fig 3B: why were only bias trials used for this analysis?

Thanks for spotting this error. The analysis reported in Figure 3B was carried out including only *neutral* trials. Comparable analyses including *biased* trials are reported in Figure 5D.

I204: "more direct measure, of" -> "more direct, measure of"

Typo corrected.

I329: "identifying the variability condition but otherwise" - "were" missing

Typo corrected.

Fig S2, caption: "men" -> "mean"

Typo corrected.

Reviewer #3 (Remarks to the Author):

I find the manuscript significantly improved, with regards to additional helpful analyses (for example the sub-sampling and the sequential-sampling models) and a clearer presentation and discussion. The authors also provided reasonable answers to most of my queries and I believe that they touch on an important distinction (encoding vs integration noise) for the decision-optimality debate. There are still a few issues, which I believe that could be further clarified and discussed, in order to obtain an effective contribution for this journal. I will elaborate on these issues below, starting from the more conceptual to the technical ones.

We thank the Reviewer for the positive assessment of our work. We hope to have addressed the issues raised by the Reviewer as detailed in our responses below.

1. How does increased-variability affect RT?

The authors have responded to my query of whether participants act upon increased variability in a different way from reduced contrast, by increasing the response-criterion, so as to require more evidence samples compensating for difficulty, by stating that while such a variation would explain the increase in RT it should keep accuracy at a fixed value. This would be indeed the case if the change in criterion is perfectly calibrated to compensate for the increase in difficulty. However, it is possible that the change of criterion is partial, resulting in both an increase in RT and a reduction in accuracy. I am not saying that this is the case, and I also accept that the authors' proposal that increased variability leads to a reduced drift (due to evidence-compression caused by outlier-neglect) is a reasonable one. It would be good, however, to see this shown in the sequential-sampling simulation. As presented now, this simulation only disproves the fixed-threshold, equal-drift/increased-drift-variability model. It should be possible, however, to extend this model analysis to also contrast between the increased-criterion hypothesis and the reduced-drift one. To do this all one has to do is to include these two models (which make different assumptions on how increased variability) affects the diffusion parameters, and see via BIC-measures, which provides a better account to the data.

We thank the Reviewer for this interesting suggestion. We have extended the DDM section in the Supplementary Information to directly compare the 'increased-criterion' (threshold) and the 'reduced-drift' hypotheses (**lines 1102-1127**). In particular, we used the HDDM toolbox (Wiecki et al., 2013, *Frontiers in Neuroinformatics*) to estimate subjects' DDM parameters (i.e., drift-rate, threshold and non-decision time) separately for each condition within a Bayesian framework. The group-level posterior distributions over

parameters showed that only drift-rate changed between conditions, whereas threshold and non-decision time were stable across conditions (Fig. S5). The HDDM results are consistent with our claim that increased variability affects drift-rate and not threshold.

Here are the relevant sentences from the main text:

“...To further evaluate how our experimental manipulations affected the choice process, we fitted a hierarchical instantiation of the drift-diffusion model (Wiecki, Sofer, & Frank, 2013) to participants’ choice behaviour on neutral trials. In line with the above results, this analysis showed that the effects of contrast and variability on accuracy and RTs were captured by a change in drift-rate and not in threshold or non-decision time (**Fig. S5**).” (Results, **lines 370-375**)

One technical point here is the fit measure that is being used in model fitting and comparison. I believe that this cannot be based on accuracy scores only, because doing so gives a high weight to accuracy and neglects RT (and then it is not very surprising if the RT fit is less good); also a difference in accuracy between .95 and 1 is not the same as one between .75 and .8, due to ceiling compression effects. Rather this should be based on Log-Likelihood for a response to be correct/incorrect and within a certain time bin (this can be also done using quantile-RTs; Ratliff & McKoon, 2008). I believe that such an exercise would significantly strengthen the conclusions (whatever they are) without requiring the more complicated model, based on outlier-neglect. I agree that the latter does not need to be simulated in this paper, but I think that it could be helpful to mention it in the Discussion as a reason for the drift reduction (if this is what the model comparison shows; in case there is a tie between drift-reduction and the criterion-increase, this can also be discussed). I agree that this is secondary to the issue of why variability has a different effect on confidence and on weights given to priors, however, RT is an important aspect of the decision-process, having implications for the issue of whether participants totally discard integration noise, or only partially do so).

We appreciate the Reviewer’s concern and have complemented our analyses with the HDDM approach as described above. The HDDM considers both choice accuracy and RT and has been shown to be superior to conventional fitting approaches in a range of scenarios (Ratcliff & Childers, 2015, *Decision*).

2. Encoding- vs. integration-noise and limited capacity.

I am still confused about how we should understand the integration noise and its relation to capacity limitations. The authors stated in response to my query that the capacity limitation does not involve a fixed number of items, but rather the degree of variability. But this is exactly what we need to explain. It can help to make this more concrete by

considering a mechanism-integration example. Consider, the LIP-accumulators, which were proposed to carry out evidence-integration in decision tasks (Gold, Shadlen, et al). The way we typically understand such a mechanism is that its advantage is exactly to do away with the capacity limitation of the WM-system. This is because one only keeps a single (or two) integrated tally (ies) of the evidence, rather than all the samples which would exceed the capacity limitation of the WM system. So, the question that would be helpful to discuss is, what do these results imply about such an integration mechanism? Few options one may consider is that the mechanism is subject to some deficiency (blinks, or recency/primacy of temporal weighting) that give rise to what is here called “integration-noise” (or perhaps that such a mechanism is not really available at all). In addition, it may be helpful to discuss whether such integration-deficiency (or noise) apply equally to temporal integration (as in the regular moving dot paradigm) or is particular to integration across space (as in this paper).

We thank the Reviewer for these interesting comments on our work and have extended the Discussion to further explain the link between integration noise, capacity limitations and temporal evidence integration. There are many higher-order sources of choice variability which would fall under our definition of integration noise: for instance, imperfect inference, information decay in working memory, temporal biases, or conflict among relevant pieces of information. Under this view, integration noise on sequential sampling tasks, such as the random dot motion paradigm, could happen at the moment of updating the ‘tally’, or when the ‘tally’ is used to inform a course of action. As such, recency and primacy effects would amount to integration noise.

Here is the relevant paragraph from the main text:

“By comparison, integration noise strictly refers to internal noise which arises at later stages of information processing, when two or more pieces of information need to be integrated either in space or time, within a limited-capacity cognitive system. There are several potential contributors to such noise. For instance, errors of inference and updating (Wyart & Koechlin, 2016), information decay in working memory (Bays & Husain, 2008), temporal biases such as recency and primacy (Cheadle et al., 2014), conflict among relevant information (Eriksen & Eriksen, 1974; MacLeod, 1991). Of course, choices may be affected by other types of noise than those considered here. For example, cognitive decisions may involve memories, sometimes distant in the past, and risk and ambiguity (Bach & Dolan, 2012; Payzan-LeNestour & Bossaerts, 2011).” (Discussion, **lines 433-440**)

Technical points

p. 3 (1st paragraph). What about optimality in the SPRT sense (Gold & Shadlen, 2001; Bogacz et al., 2006), which seems to involve temporal-integration of evidence. Should this not be included in the optimal part of the decision mechanism?

We now cite these references in the Discussion where we relate our results to temporal evidence integration – see next reply.

p. 3. Last paragraph. Why is the integration of multiple pieces of evidence difficult? (from the perspective of a LIP-integrator, it should not be?)

We now acknowledge that models of sequential integration typically assume a ‘noiseless’ integrator (as in the SPRT sense), but cite a recent influential study (Drugowitsch et al., 2016, *Neuron*) which estimated that a large fraction of choice variability is due to errors or noise in integration (**lines 443-450**), and discuss too, that as long as two or more pieces of information need to be combined, the variability between them will increase cognitive demand and thereby errors or noise in integration (**lines 433-436**).

p. 6. The bias-index shows a positive main effect of contrast (both text and D-caption). Should this not be “negative” main effect: less bias for high contrast?

This is an error on our part; we have now corrected it.

p. 9. Fig. 4. It may be helpful to explain why the noise-blind functions appear more skewed (I assume this is due to the fact that the noise is convolved with the red-blue truncated distributions in panel-A).

The Reviewer’s interpretation is correct; we now make this point in the figure caption.

p. 10, I find the correlation-data in Fig-4 A, less significant (and more difficult to follow) for the argument than the results presented in the Supplement (S2) about the overconfidence. So, you may like to consider if it is not better to swap those.

We decided to keep scatterplots as they provide a more accurate representation of the underlying data – the ‘overconfidence’ results can also be seen from the scatterplots.

p. 11. Line 306. “Models differ”. (which models?)

We now clarify that *all three* models differ with respect to predicted confidence.

p. 11 Line 329. "but otherwise aware". (are aware?)

Typo corrected.

p. 29. Sequential-Sampling Suppl. What type of drift variability is used. Is this variability between trials (or within a trial)?

We have now clarified in the text that the variability in drift is within-trial.

Reviewers' Comments:

Reviewer #1:

Remarks to the Author:

The authors have addressed to satisfaction my remaining comments.

Reviewer #2:

Remarks to the Author:

The authors addressed all my previous concerns. I appreciate them including the "ensemble" model, and showing (as expected) that it doesn't reproduce all the features of the data. I only have a few minor comments concerning the Supplementary Information:

Eqs. (7) and (8) are inconsistent with the preceding text. They put both means at +2 (rather than the +/-3 in the text above).

l963: "determined response accuracy" - "the" missing?

Eqs. (9) and (11): the A's and I's should be boldface for consistency (also in the text that follows the equations).

Reviewer #3:

Remarks to the Author:

I find the revised manuscript improved (both in its more clear presentation and with regards to the novel drift-diffusion model fits). Also the authors provided very reasonable replies to my last questions and I find the changes totally satisfactory. Thus I have no further comments and I am happy to recommend publication.

REVIEWERS' COMMENTS:

Reviewer #1 (Remarks to the Author):

The authors have addressed to satisfaction my remaining comments.

Reviewer #2 (Remarks to the Author):

The authors addressed all my previous concerns. I appreciate them including the "ensemble" model, and showing (as expected) that it doesn't reproduce all the features of the data. I only have a few minor comments concerning the Supplementary Information:

Eqs. (7) and (8) are inconsistent with the preceding text. They put both means at +2 (rather than the +/- 3 in the text above).

We have now corrected this mistake.

l963: "determined response accuracy" - "the" missing?

We believe the sentence (two lines above equation 9 circa line 1012) is grammatically correct as is.

Eqs. (9) and (11): the A's and I's should be boldface for consistency (also in the text that follows the equations).

We have now corrected this.

Reviewer #3 (Remarks to the Author):

I find the revised manuscript improved (both in its more clear presentation and with regards to the novel drift-diffusion model fits). Also the authors provided very reasonable replies to my last questions and I find the changes totally satisfactory. Thus I have no further comments and I am happy to recommend publication.